# Antitumor Potential of Marine and Freshwater Lectins

**DOI:** 10.3390/md18010011

**Published:** 2019-12-21

**Authors:** Elena Catanzaro, Cinzia Calcabrini, Anupam Bishayee, Carmela Fimognari

**Affiliations:** 1Department for Life Quality Studies, Alma Mater Studiorum-Università di Bologna, Corso d’Augusto 237, 47921 Rimini, Italy; elena.catanzaro2@unibo.it (E.C.); cinzia.calcabrini@unibo.it (C.C.); 2Lake Erie College of Osteopathic Medicine, Bradenton, FL 34211, USA

**Keywords:** marine lectins, cancer, cancer therapy, in vitro studies, in vivo studies, natural products

## Abstract

Often, even the most effective antineoplastic drugs currently used in clinic do not efficiently allow complete healing due to the related toxicity. The reason for the toxicity lies in the lack of selectivity for cancer cells of the vast majority of anticancer agents. Thus, the need for new potent anticancer compounds characterized by a better toxicological profile is compelling. Lectins belong to a particular class of non-immunogenic glycoproteins and have the characteristics to selectively bind specific sugar sequences on the surface of cells. This property is exploited to exclusively bind cancer cells and exert antitumor activity through the induction of different forms of regulated cell death and the inhibition of cancer cell proliferation. Thanks to the extraordinary biodiversity, marine environments represent a unique source of active natural compounds with anticancer potential. Several marine and freshwater organisms, ranging from the simplest alga to the most complex vertebrate, are amazingly enriched in these proteins. Remarkably, all studies gathered in this review show the impressive anticancer effect of each studied marine lectin combined with irrelevant toxicity in vitro and in vivo and pave the way to design clinical trials to assess the real antineoplastic potential of these promising proteins. It provides a concise and precise description of the experimental results, their interpretation as well as the experimental conclusions that can be drawn.

## 1. Introduction

Water covers 71% of the surface of our planet and hosts vibrant biodiversity. More than 2 million different species inhabit the seawater, many of which still await discovery [1]. Terrestrial natural compounds have always represented a source of biologically active molecules, while the marine environment is still a step behind. So far, the World Register of Marine Species checked and classified only 232,297 different species [2]. This number could be considered modest, but it mirrors how late oceans have started to be studied compared to land environments. Indeed, the spread of terrestrial natural history began during the 17th century, and escalated through the scientific travels of 18th and 19th centuries—including Charles Darwin’s expeditions—until nowadays. On the contrary, it was not until the 20th century that ocean exploration started. It should be enough to underline that before the last century, technology was not sufficient to explore deep waters [3]. However, the number of discoveries is now exponentially increasing [4].

From the perspective of a pharmacologist, the marine biodiversity represents the ideal environment to quest for new substances with therapeutic potential. Indeed, since oceans give a home to many sessile or limited-mobility species, it is logical to assume that they synthesize metabolites to protect themselves and that those metabolites can be exploited for therapeutic uses [5]. Especially in anticancer research, the marine environment is an actual gold mine. By 2017, Food and Drug Administration of the United States approved seven marine-derived pharmaceutical drugs as anticancer agents, while many other compounds are currently being studied in different phases of clinical trials in oncology and hematology fields (4 in phase III, 6 in phase II, 8 in phase I) [6]. It has been estimated that 594,232 novel compounds of marine origin are waiting to be studied, assuming that between 55 and 214 could be approved as new anticancer drugs [4].

Nonetheless, if the natural world is so rich in bioactive molecules, why is the cure for cancer one of the hardest challenges scientists are facing for the last two centuries? First of all, it is incorrect to refer to cancer as a single disease. At least 200 different types of tumors exist. For instance, the organ where the tumor originates gives rise to specific neoplastic lesions; thus, breast cancer is different from lung cancer. Likewise, the type of cells from which cancer derives is crucial, and carcinomas derived from epithelial cells are different from lymphomas that derive from cells of the immune system. Furthermore, cancer is not quiescent, and the possibility to evolve and acquiring resistance to a full-blown effective therapy is very high [7]. Thus, it is pretty unlikely finding a single cure for all cancer types. Moreover, most of the anticancer agents used in clinic cause unacceptable toxicity that limits their compliance and effectiveness, hampering the positive outcome of cancer treatment. Since the cytotoxic effect of the antineoplastic drug also affects healthy cells, the primary cause of toxicity is the lack of selectivity towards tumors cells [8,9,10]. Thus, an interesting antitumor strategy would foresee a drug able to eradicate tumor cells distinguishing between them and healthy cells.

Therefore, the next question is: how do cancerous cells differ from normal ones? What all neoplastic cells have in common and that distinguish them from normal cells is the acquisition of different abilities. These abilities allow them to proliferate persistently, resist to cell death, circumvent growth restraint stimuli, reprogram energy metabolism, and escape the immune system recognition [11]. All the pathways involved in the acquisition of these capabilities make tumor cells unique and ideally can be exploited to obtain a therapy that selectively kills cancer cells. Targeting fast proliferating cells was one of the first strategies designed for this aim. However, some healthy tissues have a high physiological rate of proliferation and damage to normal cells cannot be avoided [12]. More, to survive in hostile environments, cancer cells deregulate cellular energetics. Indeed, reactive oxygen species (ROS) production is increased while the antioxidant machinery is often impaired. Thus, the basal oxidative stress level of cancer cells is higher than that of normal cells. By the consequence, enhancing ROS levels in cancer cells most likely provokes the selective eradication of these cells rather than normal cells [8,10,13]. In addition to oxidative stress, many other pathways, such as increased glycolysis (the Warburg effect), hypoxia and acidity have been targeted [9,11]. Nonetheless, hitting any one of these targets does not lead to a complete selectivity of action. Conversely, an effective strategy to identify and selectively hit cancer cells could be aiming at the phenotypic changes that come along with the malignancy transformation, such as the glycosylation pattern at the cell surface. In normal conditions, most proteins and lipids of eukaryotic and prokaryotic cells are coated by a sugar structure called glycocalyx. Glycocalyx plays a pivotal role in cell-cell recognition, communication, and intercellular adhesion. It is composed by mono-, oligo- or polysaccharides and changes in a predictable fashion throughout normal cell development. Glycosyltransferase enzymes, located in the Golgi apparatus, catalyze the construction of this carbohydrate structure. During malignant transformation, alterations in glycosyltransferases and their gene expression occur [14], and glycosylation profile of cells is irreversibly altered, conferring unique phenotypes to cancer cells [15]. For instance, protein fucosylation and sialylation, i.e., the addition of fucose moieties or sialic acid to proteins, is a common event in many cancer types [16]. The modifications of cell surface sugar patterns and in particular glycoproteins can be exploited to target cancer cells exclusively. Lectins are non-immunogenic glycoproteins that specifically bind unique carbohydrates residues on cell surfaces. Thus, they represent an attractive tool to detect and kill cancer cells [17]. 

Several studies have shown that plant and animal lectins exert different biological activities. These compounds inhibit bacterial [18] and fungal growth [19], and have immunomodulatory properties [20,21]. On cancer cell lines, they exhibit antiproliferative properties [20,22], promote apoptosis and autophagy, and block angiogenesis [20,22]. Moreover, their ability to bind to carbohydrates could make them useful for the detection of subtle modifications of carbohydrate composition, which appears during cancer transformation [23]. However, although anticancer activity of terrestrial lectins has been summarized in several reviews [20,24,25], to our knowledge no comprehensive report highlighting uniquely the antineoplastic activity of marine lectins has been written.

This review will explore the antitumor activity of lectins from both marine and freshwater organisms. A short overview of lectins will be followed by the presentation of the most studied lectins that showed overt anticancer potential. Diagnostic applications of lectins will not be discussed since it goes beyond the aim of this review.

## 2. Lectins

Lectins are expressed by animals, plants, fungi, and bacteria. They contain a domain for carbohydrate recognition that allows them to bind with carbohydrates without modification, i.e., they do not exhibit enzymatic activity on carbohydrate molecules. Since they reversibly and specifically bind carbohydrates, whether simple sugars or complex carbohydrates, lectins are capable of precipitating polysaccharides and glycoproteins or agglutinating cells [17].

Lectins can be classified using different criteria, including cellular localization, structural and evolutionary sequence similarities, taxonomic origin, carbohydrate recognition, function, and structure [26] (Figure 1).

Lectins play a pivotal role in cell-cell and host-pathogen interaction and communication, tissue development, sugar storage, and other mechanisms of cell survival and immune system stimulation. All endogenous lectins can be involved in both physiologic and pathologic processes [25]. Based on their characteristics, they can induce apoptosis and autophagy and inhibit angiogenesis, thus representing an interesting anticancer strategy [25]. Their mechanism of action consists of recognizing and reversibly binding specific carbohydrate moieties on cell surfaces. In particular, lectins bind to a specific sequence of monosaccharide moieties within glycosylated proteins, lipids, and glycans that are on cell surface. In humans, several types of cells express these proteins, from epithelial cells to antigen presenting ones and, depending on the type of lectin, they can be exposed on the cellular membrane or secreted into the extracellular matrix [27]. Each lectin usually has more than one binding site for the sugar units. As a consequence, they are able to bind different carbohydrates exposed on the surface of different cells, thus enabling cell-cell interaction [22]. Unique glycosylated-proteins expressed in tumor cells become the target of lectins that bind the characteristic glycan-chains. This process is the key to discriminate abnormal glycosylation moieties on the surface of cancer cell membranes [28]. Not all cancer types show the same glycosylation pattern, but they share common features. In particular, during the tumorigenesis, specific subsets of glycans on the cancer cell surface face different modifications, such as enrichment or decrease in their main components. The most recurrent changes on cell surface glycosylation patterns are enrichment in sialylation, branched-glycan structures, and the generation of the so-called “fucosylation core” [29,30]. For instance, the tetrasaccharide carbohydrate moiety sialyl-LewisX and sialyl-Tn antigen represent well-known markers of malignancies, since they are found in almost all cancer types [31]. Besides, O-linked and N-linked glycosylation create the most common branches on the tumor cell surface glycocalyx. For example, O-linked glycosylation is promoted by the entry of a GalNAc sequence, which can then be expanded with the addition of other different structures. Simultaneously, different blocks of sugars can be attached to the main branch through an amide group of an asparagine residue resulting in N-linked glycosylation. Among others, ovarian, skin, breast, and liver cancers show these types of modifications [30,32]. To complete the picture, core fucosylation, i.e., the enrichment in α1,6-fucose of the innermost GlcNAc residue of N-glycans, is a specific trait in many cancers, such as lung and breast cancer [29,33].

The recognition of glycosylated proteins on the outer cellular membrane represents only the first step necessary to trigger cancer cell death. After the binding with the carbohydrate moiety, lectins can be internalized, and there, in the intracellular environment, they trigger specific pathways that lead to cell death. Furthermore, taking these notions into consideration, exogenously induced expression of lectin inside the tumor cells could be another effective strategy to promote tumor eradication. For this reason, vaccinia viruses (VVs) have been exploited as replicating vectors harboring the genes needed for lectin expression and thus as an efficient way to transport lectins into cancer cells [34].

Marine and freshwater lectins show anticancer properties in vitro and in vivo thanks to the binding of cancer cell membranes, causing cytotoxicity, apoptosis, other forms of regulated cell death, and inhibition of tumor growth. A literature review was performed to identify publications on the antitumor potential of lectins isolated from water organisms such as invertebrates, vertebrates and tunicates, fish, and algae. The following paragraphs will present their activity as direct antitumor agents or through their delivery via VV.

## 3. Lectins from Marine and Freshwater Algae

Aquatic algae are a group of simple organisms containing chlorophyll. To sustain themselves, they almost only need light to carry out photosynthesis. They inhabit seas, rivers, lakes, soils, and grow on animal and plants as symbiotic partners. The complexity of these organisms ranges from the simplest unicellular to pluricellular entities that reorganize themselves to form simple tissues [28]. At least 44,000 species of algae have been identified and named [29], and among them 250 have been reported to contain lectins [3]. However, only a few of them have been adequately studied to investigate their antitumor potential (Table 1). For some of them, the only cytotoxic potential has been evaluated.

The most characterized lectin of algae origin, the *Eucheuma serra* agglutinin (ESA), was extracted from the homonymous red macroalga. ESA is a mannose-binding lectin able to promote apoptosis on different cell lines and animal tumor models (Table 1). ESA amino acid structure is composed by four tandemly repeated motifs, each of them representing one binding site for mannose sequence. Specifically, each repeated motif binds specifically high mannose N-glycans with a minimum dimension of tetra- or penta-saccharide, such as Man(alpha1-3)Man(alpha1-6)Man(beta1-4)GlcNAc(beta1-4) GlcNAc [43].

ESA promotes cell death of many cancer cell lines, such as Colo201 (human colon adenocarcinoma), Colon26 (murine colon-carcinoma), HeLa (human cervix adenocarcinoma), MCF-7 (human breast adenocarcinoma), OST (human osteosarcoma), LM8 (murine osteosarcoma) [36,37,38,39]. In each of these cell lines, the mechanism of cell death is apoptosis, as demonstrated by DNA fragmentation, exposition of phosphatidylserine, and activation of caspase-3. The activity of ESA is tumor-type dependent and, comparing distinct studies, cervix adenocarcinoma and colon adenocarcinoma came back the most sensitive followed by osteosarcomas and breast cancer, which respond to a higher concentration of this lectin [36,37,38]. Since lectins, in general, have precise targets, this behavior probably reflects the different glycosylation pattern of different types of tumor. Certainly, the different glycosylation pattern between normal and cancer cells is behind the lack of ESA activity on non-transformed cells. Indeed, ESA did not affect the viability of normal fibroblasts and the non-tumorigenic epithelial MCF10-2A cell line [37,38]. Furthermore, their selectivity for tumor cells translates into lack of toxicity in vivo. For instance, ESA delayed the growth of Colon26 cells injected on BALB/c mice without affecting the body weight nor causing direct death, thus showing promising in vivo tolerability [36].

The ability of lectins to selectively target cancer cells can be exploited not merely to kill tumors, but also to deliver antitumor drugs on cancer cells. Furthermore, the antitumor activity of lectins could theoretically sum to that of the antitumor drug. With this aim, the development of a selective drug delivery system (DDS) was designed, and lipid vesicles resembling a microcapsule were tagged with ESA [38]. The microcapsules were made by sorbitan monooleate (Span80) with or without poly(ethylene glycol) (PEG) [37,39]. PEG was added in order to prolong the half-life of the vesicles compared to normal liposomes since PEGylation should decrease the reticuloendothelial uptake. First of all, it was demonstrated that both ESA-labelled DDSs target the exact same carbohydrate-sequence of free ESA, and that the drug transportation system does not abolish its cytotoxic activity, nor selectivity towards tumor cells [37,44]. ESA-immobilized lipid vesicles reached and bonded Colo201, HB4C5, OST, while no interaction was recorded with normal human fibroblasts and MCF10-2A [37,38,39] (Table 1). Indeed, all microcapsules directly exhibited pro-apoptotic activity on Colo201 and OST cells [37,39], while little effect was observed in normal MCF10-2A [39]. In vivo, the injection of the vesicles delayed tumor growth in nude mice bearing Colo201-derived tumors [39] (Table 1). No study directly compared the antitumor activity of the PEGylated vs. the not-PEGylated ESA-vesicles, while only in vivo bioavailability experiments were performed, showing no difference between the two systems [37,39]. In fact, after the injection of one or the other radioactive-labeled microcapsules in nude mice bearing Colo201 tumor cells, no difference in tumor accumulation nor uptake was recorded [39]. Finally, to understand if the DDS was able to transfer the encapsulated material into tumor cells, two different research groups [38,39] loaded them with the fluorophore fluorescein isothiocyanate (FITC) and treated cells with this complex. They demonstrated that FITC is internalized into tumor cells cytoplasm, both in vitro and in vivo, and showed the interesting potential of ESA-vesicles as a carrier of antitumor drugs as well as direct antineoplastic agents [38,39].

If ESA was not particularly active on MCF-7, two isolectins isolated from *Solieria filiformis* (Sfl) proved their ability to fight breast cancer (Table 1). Sfl-1 and Sfl-2 are mannopentose-binding lectins differing from each other for only 18% of the aminoacidic sequence. SfL-1 and SfL-2 consist of 268 amino acids in the form of four identical tandem-repeat protein domains. They both are constituted by two β-barrel-like domains made of five antiparallel β-strands, which are connected by a short peptide linker [41]. Despite the different biological activity of Sfls and ESA, both Sfl isoforms share at least 50% of their amino acid sequence with it [45] (Figure 2A). Thus, we can hypothesize that the similar structure allows them to both bind the same glycan structure, while the remaining part is responsible for the different cytotoxic activity.

Both Sfls isoforms promote MCF-7 cell death to an equal extent, while they were not able to entirely kill primary human dermal fibroblasts. However, at a higher concentration than those promoting cell death, Sfls induce fibroblast proliferation [41]. Keeping these data in mind, the difference in the sensitivity of cancer and normal cells allows the identification of an interval of concentration toxic for the tumor ones. The mechanism of anticancer action of these lectins lies in the induction of apoptosis. Normally, apoptosis can be promoted by triggering the intrinsic and/or extrinsic pathway. The first one involves the B-cell lymphoma 2 (Bcl-2) protein family, mitochondrial permeability, and the creation of the apoptosome. The extrinsic pathway involves death receptors, death-inducing signaling complex and caspase activation. The two pathways share effector caspases. Due to congenital or acquired genetic alterations of cancer cells, one or, in the worst case, both pathways are often compromised [46]. Thus, the ability to activate both pathways maximizes the probability to promote programmed cell death successfully and minimizes the one to acquire resistance. Sfls upregulate the gene expression of the effector caspase-3 and trigger both the intrinsic [upregulation of the proapoptotic gene Bcl-2 associated X protein (Bax); downregulation of the anti-apoptotic gene Bcl-2; increased expression of caspase-9] and the extrinsic (increased expression of caspase-8) apoptotic pathways, qualifying themselves as very interesting anticancer agents. Further studies will be needed to understand if and how the 18% difference between the two isoforms affects their biological properties [41].

*Ulva pertusa* lectin 1 (UPL1) is an N-acetyl d-glucosamine-binding lectin that interacts with several intracellular pathways involved in proliferation and cell survival (Table 1). UPL1 primary structure does not share amino acid sequence similarity with any known plant or animal lectin. cDNA sequencing analysis showed that UPL1 is 1084 base-pair long and encodes for a premature protein of 203 amino acids. During post-translational modifications, 53 amino acids are lost and the N-terminal sequence of UPL1 starts at amino acid number 54 [47]. Intracellular delivery of UPL1 through a flag adenovirus does not exert an antiproliferative effect on the human papillomavirus-related endocervical adenocarcinoma BEL-7404, and hepatocyte-derived cellular carcinoma Huh7 cells. UPL1 mediates both the activation of the extracellular-signal-regulated kinase (ERK)-mitogen-activated protein kinase (MAPK), also known as MAPK/ERK kinase (MEK), signal and the phosphorylation of p38 MAPK. Since constitutive activation of ERK promotes cell survival and drug-resistance [47] and p38 MAPK can act as a survival factor, this could be the cause of UPL1 inability to stop cancer growth [42]. Indeed, the inhibition of each one of these pathways enhanced the antiproliferation activity of exogenous UPL1. Even if the cellular functions of tubulin in cancer is not clear [48], the ability of UPL1 to modulate MAPK pathways lies on its binding with β-tubulin. Probably, this critical protein is glycosylated in a specific manner providing recognition sites for the lectin. What is certain is that when the binding UPL1-tubulin was impeded, nor ERK-MEK or p38 MAPK were modulated, and UPL1 was able to kill cancer cells [42]. Moreover, only on Huh7, UPL1 triggered autophagy. If autophagy-associated cell death is considered a tumor suppressor characteristic, it can also enhance the survival of neoplastic cells exposed to metabolic stress and foster metastasis [49]. As a consequence, attention must be paid to assessing UPL1 real antitumor potential, alone or combined with survival signaling inhibitors, such as MAPK inhibitors [42].

Since lectins are such attractive pharmaceutical agents, the identification of new organisms able to produce them is compelling. However, the qualitative analysis of extracts could be time- and resource-consuming. Affinity electrophoresis or photometric assays are two tricky techniques used to carry out this task [48]. Additionally, an easy and fast method for understanding if a substance contains lectins is to evaluate its hemagglutination activity. Anam et al. [35] studied different crude extracts of red macroalgae (*Nitophylium punctatum, Acanthophora spicifera, Acrocystis nana, Helminthora divaricata* and *Gloiocladia repens*) and showed different degrees of hemagglutination activity. This activity, however, did not reflect the potency of their antitumor potential. All five crude extracts were able to kill MCF-7 and HeLa cells to a different extent (Table 1). They were all tested at a concentration of 100 µg/mL. *Acrocystis nana* extract was the second extract for hemagglutination activity, but the most potent in killing HeLa cells, inhibiting almost 50% of cell growth after 24 h. For MCF-7 cells, the extract more enriched in lectins (*Gloiocladia repens)* was the most cytotoxic with inhibition of almost 30% cell growth. All other extracts affected cell growth in an interval ranging between 2 and 15% [35]. These data, although not outstanding, may be deemed a starting point for the complete purification of the extracts in order to isolate the lectins. In general, consequential chromatographic steps are needed to purify lectins from crude extracts, such as affinity chromatography, ion-exchange chromatography, hydrophobic chromatography, and gel filtration chromatography [17]. In this case, the complete purification of the lectins from the crude extracts will clarify if lectins alone still have anticancer potential and whether that potential is higher than that of the extract. Anam et al. [35], who authored the study, suggest that extracts could contain impurities that dilute the samples and that purification should allow increasing the antitumor activity [35]. However, very often, extracts are more potent than the single molecules alone [49]. By the consequences, it will be interesting to see how these aspects pan out.

## 4. Lectins from Marine and Freshwater Invertebrates

Animals belonging to marine invertebrates have been exploited for their biological activity since the times of ancient Greece. Hippocrates and Galen, two pioneers of modern medicine, extensively narrated about dietary and pharmaceutical uses of shellfish, sponges or cephalopods, and their prescriptions containing marine invertebrates along with other ingredients have been found. Marine invertebrates consist of a large variety of organisms and have been categorized into over 30 different *phyla* [50]. They do not have an innate immune system nor develop an adaptive immune response against pathogens. Thus, as an immune defense they constitutively express small peptides that are induced upon danger is sensed. Lectins are among these proteins [51], and interestingly, they showed impressive antitumor activity both in vivo and in vitro (Table 2).

### 4.1. Phylum Mollusca and Arthropoda 

The phylum Mollusca represents the second biggest *phylum* on earth. Approximately 90% of mollusks fall into the Gastropoda class, followed by Bivalves and Cephalopods [70]. Shelled mollusks are the most widely used in traditional medicine and together with the crustaceous (*phylum* Arthropoda) fall into the shellfish family [71]. Several studies revealed that the richness in hepato-pancreas mass, the characteristic open circulatory system, the filtering abilities and the shell arrangement make bivalves and crustaceans a remarkable source of molecules [72], such as the unique lectin proteins that can be exploited as therapeutic agents.

*Crenomytilus grayanus* lectin (CGL) is an intriguing lectin isolated from the homonymous bivalve belonging to the Mytilidae family. It consists of three highly similar tandem sequences of amino acids for a total of 150 residues. Secondary structure envisages a predominance of β-structure that sometimes alternates with α-ones [73]. CGL has a particular structure that makes it able to specifically recognize the globotriaosylceramide (Gb3) resulting in Gb3-dependent cytotoxicity. Gb3 is a globoside, and thus a non-protein cluster of differentiation composed by a galactose-lactosylceramide sequence [74]. It is overexpressed in several human tumors with intrinsic or acquired multidrug resistance [75]. CGL blocks cell proliferation and promotes cell death on Gb3-expressing tumor cells, such as Raji cells (Burkitt’s lymphoma) and in a lesser extent MCF-7 (breast carcinoma) (Table 2). No effect was recorded on K562, a human myelogenous leukemia cell line not expressing Gb3 [53,54]. Furthermore, on Raji cells, the CGL cytotoxicity was completely hindered when Gb3 was enzymatically degraded [54]. On the same cell line, CGL promoted a G_2_/M cell-cycle arrest that led to apoptosis, as demonstrated by phosphatidylserine externalization, activation of caspases-3 and caspase-9, and poly (ADP-ribose) polymerase (PARP) cleavage [54]. Even though further studies are needed to confirm the following assumption, the high expression of Gb3 on some cancer cells and the very specific action of CGL for these cells bodes well for a selective activity of this lectin towards neoplastic lesions and let us presume that it would bring to lack of toxicity in vivo.

*Mytilus galloprovincialis*, better known as Mediterranean mussel, contains a lectin, Mytilec, that shares a similar behavior with CGL. No wonder, they share 50% in amino acid sequence and bind the same glycan moiety [45] (Figure 2B; Table 2). Mytilec is incorporated into cells through the interaction with Gb3 and, thanks to that, it promotes cytotoxic effects. As CGL, it exhibits antitumor activity only on Gb3-expressing cell lines, such as Raji’s and Ramos, another Burkitt’s lymphoma cell line (Table 2), while no effect was induced on K562 [35,36]. The mechanism of action of Mytilec has been investigated in Ramos cells, where it promotes apoptosis and activates all MAPK pathways. MAPK system consists of three sequentially activated protein kinases that play a central role on different transduction pathways. These pathways are involved in processes, such as cell proliferation, differentiation, and cell death in eukaryotes [76,77]. Precise extracellular stimuli elicit the phosphorylation cascade and provoke the activation of a MAPK by the consecutive activation of a MAPK kinase kinase (MAPKKK) and a MAPK kinase (MAPKK). Briefly, after the stimulus is inferred, MAPKKK phosphorylates and triggers MAPKK, which, in turn, activates MAPK. Here, MAPK phosphorylates different substrates in the nucleus and cytoplasm, inducing variations in protein function and gene expression that deliver the proper biological response. Depending on which stimulus triggered the MAPK system in the first place and the intended biological effect, three leading families of MAPK can be activated. Growth factors and mitogens activate ERK, which controls cell survival, growth, differentiation and development; stress, inflammatory cytokines and, once more, growth factors prompt c-Jun *N*-terminal kinases (JNKs) and p38. These two latter MAPKs, in turn, deal with inflammatory response, apoptosis, cell growth, and differentiation [78]. Mytilec activates all these three pathways, but only in Gb3-expressing cell lines. The activation of the ERK pathway increased the levels of tumor necrosis factor-α (TNF-α) and the cell-cycle inhibitor p21, but it was excluded to be the cause of apoptosis induction. The role of JNK and p38 on the pro-apoptotic properties of Mytilic has still to be verified [60].

The gonads of the adult giant marine *Aplysia*, a sea slug belonging to Gastropoda class, contains the corresponding lectin: *Aplysia gonad* lectin (AGT). It is a Ca^2+^-dependent lectin and is composed of two 32-33 kDa subunits [79]. Although the majority of studies about AGT describe its chemical characteristics, this galacturonic acid-galactose-lactose-binding lectin has been injected on mice and it was found able to abolish tumor appearance when tested at 30 µg and decrease lung tumor burden of 60% when tested at 5 µg, compared to untreated animals [79,80]. However, no information is available on treatment time and route of administration [80].

iNol is a very big lectin physiologically synthesized by the Arthropoda slipper lobster (*Ibacus novemdentatus*) probably to destroy pathogens. iNol consists of five subunits, which are composed by 70-, 40-, or 30-kDa polypeptides that are held together by disulfide bonds. Under physiological conditions, it has a polygonal structure. Despite the high molecular weight, this lectin is incorporated into mammalian cells, such as HeLa, through endocytosis thanks to the binding with N-acetylated glycan moieties on the cell surface. iNol kills HeLa, MCF-7, T47D (breast cancer) and Caco-2 (colon cancer) cells (Table 2). HeLa cells were the most sensitive to iNol activity, so they were used to understand its apoptotic potential. iNol promotes DNA fragmentation and activation of caspase-3 and caspase-9 only if the lectin is able to bind its carbohydrate-ligand, demonstrating that its antitumor potential is strictly linked to its lectin nature [58].

Chinese or Japanese horseshoe crab or tri-spine horseshoe crab are the common names of *Tachypleus tridentatus*, an Arthropoda resembling a crab, but more closely related to spiders and scorpions. Despite the not gracious description, *Tachypleus tridentatus* represents a source of several lectins, among which the rhamnose-binding Tachypleus tridentatus lectin (TTL). TTL characterization has not been completely clarified. Different studies agree on the fact that this lectin is a multimer, but it is not clear if it is composted by hexamers and octamers [81] or tetramers [82]. The antitumor activity of TTL was assessed in vivo, after its genomic insertion into an oncolytic VV (oncoVV-TTL). Balb/c nude mice were subcutaneously engrafted with MHCC97-H liver cancer cells and then treated with oncoVV-TTL. OncoVV-TTL was able to replicate inside tumor cells and significantly reduced tumor growth compared to onvoVV-only treated mice [55] (Table 2).

Another replication-deficient adenovirus was modified to encode the gene of *Haliotis discus discus* sialic acid-binding lectin (Ad.FLAG-HddSBL). HddSBL is an another Ca^2+^-dependent binding lectin. It has 151 amino acid residues, but so far the glycosylation site has not been identified yet [83].

Ad.FLAG-HddSBL promoted cell death of hepatocellular carcinoma (Hep3B), lung cancer (A549 and H1299), and colorectal carcinoma (SW480) in a tumor-dependent manner, as the different sensitivity of these cell lines to this compound suggests [56]. Hep3B resulted in the most sensitive cell line to Ad.FLAG-HddSBL, followed by A549, H1299 and last SW480 (Table 2). Yang et al. [56], who authored the study, suggested that the different intracellular metabolism of sialic acids is the cause of the different activity of Ad.FLAG-HddSBL. On Hep3B, Ad.FLAG-HddSBL’s mechanism of action has been investigated. It promotes a regulated form of cell death, since it triggered the mobilization of annexin-V to the outer cellular membrane and decreased the expression of the anti-apoptotic factors Bcl-2 and X-linked inhibitor of apoptosis protein (XIAP). Curiously, it did not affect PARP expression, and thereby it does not activate caspases at the analyzed time point [56]. Then, two possibilities arise: either it activates PARP at a different time point, or it promoted one of the so-called non-canonical cell deaths, which are caspase-independent forms of regulated cell death.

### 4.2. Phylum Porifera

The name Porifera means “pore bearer”, and it is the most characteristic feature of sponges. Indeed, they are invertebrates sessile characterized by the lack of digestive, nervous, circulatory, and immune system. To get food, oxygen, and to reject all discards, they maintain a continuous water circulation in their body. Sponges do not have a physical barrier that protects them from predators, and the only way to survive is to produce secondary toxic metabolites. Furthermore, Porifera *phylum* is characterized by a unique biodiversity that results in a vast diversity of metabolites. Several studies have shown the chemopreventive and antitumor properties of many of them [5].

Two lectins originated from different species of the genus *Haliclona* showed an outstanding antitumor potential. *Haliclona cratera* lectin (HCL) was extracted, purified and tested for antitumor activity on human cervical and melanoma cell lines. It displayed cytotoxicity on both tumor models, which had almost the same sensitivity (half maximal inhibitory concentration 9 µg/mL vs. 11 µg/mL, respectively) [67] (Table 2). Another species, *Haliclona caerulea*, produces a peculiar lectin called halilectin-3 (H3). Its primary structure shows no similarity to any other animal lectin and consists of 251 amino acids, of which 145 arrange themselves into an α-chain and 106 into two β-chains. Alongside, quaternary structure has a heterotrimer conformation stabilized by disulfide bonds [84], which differs from the dimeric [67], trimeric [85], tetrameric [86] or multimeric [87] structure of all other lectin originated from sponges characterized so far. H3 is able to promote cancer cell death exploiting different mechanisms. On MCF-7 cells, H3 triggers both intrinsic and extrinsic pathways of apoptosis promoting an early upregulation of p53 and inhibits cell proliferation causing an accumulation of cells in the G1 phase. In addition to apoptosis, autophagy-associated cell death has been recorded after a few hours of MCF-7 treatment with H3. In particular, H3 upregulated the microtubule-associated protein 1A/1B-light chain 3 (LC3) expression, promoting the accumulation of LC3-II (Table 2). Contextually, autophagosome vesicles have been identified microscopically [66]. The authors of the study [40] hypothesize that both apoptosis and autophagy can be triggered thanks to Bcl-2 downregulation or p53 upregulation. To complete the picture, H3 has been found able to promote anoikis, a particular form of cell death that happens when cells are stimulated to detach from the extracellular matrix [66]. In normal conditions, cells need to adhere to the tissue where they grow since the essential growth factors and other survival signals are provided by the extracellular matrix and proximal cells [88]. H3 reduced the expression of integrin α6β1 and interacted with integrin α5β1, the fibronectin receptor, thus impairing MCF-7 adhesion and promoting cell death [66]. The involvement of apoptosis, autophagy, and anoikis in the cytotoxicity of H3 makes it a very promising antitumor lectin, able to fight cancer on several fronts and thus reduce drug resistance.

A lactose-binding lectin isolated from the sponge *Cinachyrella apion* (Cal) exhibited antitumor activity on human cervical (HeLa) and prostate adenocarcinoma (PC-3) cells, while milder cytotoxicity was recorded on non-transformed 3T3 mouse fibroblasts (Table 2). Cal primary structure consists of eight subunits of 15.5 kDa assembled by hydrophobic interactions and does not need Ca^2+^, Mg^2+^, and Mn^2+^ ions to bind sugars [87]. On HeLa, it promoted the typic phenotype changes of early apoptosis and modulated some of the pro-apoptotic proteins of the intrinsic pathway [64]. However, the inhibition of caspase activation did not significantly reduce its cytotoxicity. This suggests that apoptosis is not the leading mechanism through which Cal promotes cytotoxicity. Alternatively, the inhibition of the apoptotic pathway could activate a different pattern of regulated cell death [89]. Further studies are needed to untangle this knot. 

Hol-18 is an *N*-acetylhexosamine-binding lectin isolated from the Japanese black sponge *Halicondria okadai*. It is a 72 kDa tetrameric lectin organized into four non-covalently bonded 18 kDa subunits [68]. Few studies demonstrated its newsworthy antitumor potential on Jurkat (acute T cell leukemia), K562 [68], HeLa, MCF-7 and T47D cells (Table 2). On Hela, MCF-7, and T47D, Hol-18 promoted apoptosis and activated the MAPK-ERK signal [69], showing once again how this pathway is often involved in marine lectin mechanism of action. No effect was recorded on Caco-2 cells since these cells were not able to internalize Hol-18 at all [69].

The recombinant oncoVV-AVL is the result of the insertion of the *Aphrocallistes vastus l*ectin (AVL) into an oncolytic VV. AVL is a 34 kDa glycoprotein with a 24 kDa proteinaceous core *i.e.,* the lectin core. It counts 191 amino acids and is a Ca^2+^-dependent lectin. It is characterized by vast hydrophobic sequences that allow it to bind the target cell membranes and facilitates the interaction with sugars [90]. The insertion of this lectin in the oncolityc VV allowed its exogenous expression into infected cells. It promoted cell death on several tumor cell types (Table 2), such colorectal cancer, glioma, and hepatocellular carcinoma cell lines. The mechanisms of cytotoxic activity of OncoVV-AVL have been investigated only on HCT116 colorectal carcinoma cells. It promoted regulated cell death but without the cleavage of caspase-8 nor the modulation of the pro-apoptotic Bax protein. Digging deeper into oncoVV-AVL mechanism of action, it has been demonstrated that ERK activation is necessary to allow virus replication. Since the virus replication is necessary to express the AVL lectin, it can be postulated that ERK is essential for oncoVV-AVL activity. OncoVV-AVL was tested also on Balb/c nude mice bearing BEL-7404- or HCT116-derived tumors. In both cases, oncoVV-AVL efficiently inhibited tumor growth [63] (Table 2).

Queiroz et al. [65] described the cytotoxic activity of the *Cliona varians* lectin (CVL) on different tumor cell lines and healthy cells. CVL is a Ca^2+^-dependent glycoprotein. It is a tetramer of 114 kDa composed by different subunits of 28 kDa linked by disulphide bridges [91]. CVL killed K562 and Jurkat cells, while all solid tumor cell models (melanoma, renal carcinoma, and prostate tumor) and normal human peripheral blood lymphocytes were found to be insensitive to CVL (Table 2). On K562, it increased p21 and repressed the expression of pRb (retinoblastoma protein), suggesting the ability to block the cell cycle. More, CVL promoted morphological changes typical of apoptosis while cytofluorimetric analysis showed a scenario composed by the presence of both primary necrotic and late apoptotic cells. No caspase-3, caspase-8 or caspase-9 activation has been recorded, but CVL stimulated the translocation of cathepsins B (CTPB) from the vesicular compartment to the cytoplasm [65]. The role of the lysosomal cysteine protease CTPB in cell death is intriguing but not yet fully understood. CTPB modulates both necrosis and regulated cell-death processes in a tumor-type fashion. In some cell lines, it acts as an initiator of the intrinsic apoptotic pathway, modulates Bcl-2 family proteins such as Bax, Bcl-2, and Bid, and activates the caspases cascade [92,93]. In others, rather than activate effector caspases, it represents a caspase mediator, triggering some of the morphological changes of apoptosis [93]. For example, the release of CTPB enhances the release of cytochrome c and the caspase activation in hepatocytes treated with TNF [93], while on fibrosarcoma cells exposed to the same stimulus CTPB operates as a downstream mediator of the caspase apoptosis cascade [94]. CTPB is also involved in a caspase-independent form of regulated cell death, i.e., necroptosis. Necroptosis shares with necrosis the morphological features but, in contrast to the latter phenomenon, defined molecular pathways drive it. Necroptosis is triggered through the activation of death receptors such as tumor necrosis factor receptor 1 (TNFR1). TNFR1 promotes the formation of the so-called necrosome, a complex formed by receptor-interacting protein kinase 1 (RIP1)–RIP3–mixed lineage kinase domain-like protein (MLKL) [95]. The complex activates downstream events such as CTPB mobilization, which actively promotes programmed necrosis [96]. In CVL-mediated activity, the ablation of CTPB restored almost entirely cell viability, while TNFR1 was upregulated and the subunit p65 of the nuclear factor kB (Nf-kB) was downregulated. The authors of the study [65] suggest that this latter event could be the cause of the increase in both the pro- and anti-apoptotic Bax and Bcl-2 proteins. Even if the Bax/Bcl-2 protein ratio is not known, the authors still suggest that it could be the cause of what they call “caspase-independent apoptosis” [65]. Taken together, these data suggest the ability of CVL to promote a CTPB-mediated programmed form of cell death. Annexin V exposure—a hallmark of the early stages of regulated cell death—the upregulation of TNFR1, and the complete absence of caspase activation recall necroptosis, but further studies are needed to assess if apoptosis, regulated necrosis, or both events trigger CVL-mediated cell death.

### 4.3. Phylum Chordata

Chordata is another marine filter-feeding phylum. They have tubular openings through which water goes in and out. They are sessile and very often fixed to rocks or similar surfaces [97]. Tunicates, in particular, represent a source of didemins, such as aplidine and trabectin, molecules with anticancer, antiviral, anti-inflammatory, and immunosuppressive potential [98,99,100]. Furthermore, they contain lectins.

The most compelling tunicates’ lectin is the N-acetyl-D-glucosamine-binding *Didemnum ternatanum* lectin (DTL). DTL is a homotrimer which activity is not dependent by Ca^2+^ and Mg^2+^. It contains relatively high amounts of Gly, Ala, Asx, Glx Leu, Val residues and, as for the other lectin isolated from marine sponges, it is composed in part by carbohydrate (around 1.3%) [101]. Since one of the most critical functions of lectins is the cell-matrix interaction, the antitumor potential of DTL on HeLa cells has been studied in different microenvironment conditions. DTL showed different behavior on tumor cells depending on anchorage status (Table 2). In anchorage-independent conditions (cells cultivated in soft agar), DTL promotes cell growth, while in normal adhesion conditions (cells cultivated in adhesion plates) it inhibits cell proliferation and triggers cell differentiation in a way that promotes cell attachment [62]. Thus, cell microenvironment plays a key role in the DTL’s activity. Taking this in mind, an interesting approach to better predict the anticancer potential of lectins could be using tumor 3D cultures built with specific bioreactors. Standard in vitro models do not capture the complex tumor biology and do not consider cell-to-cell and cell-to-matrix interactions. Perfusion-based bioreactor systems create heterogeneous cell populations, and an optimal physiological cell-cell and cell-extracellular matrix interactions, perfectly miming tumor microenvironment [102,103].

## 5. Lectins from Marine and Freshwater Vertebrates

### 5.1. Amphibians

Amphibians are ectothermic, tetrapod vertebrates that count several species worldwide, except Antarctica [104]. Behavioral as well as biochemical, physiological, and morphological adaptations make them able to survive in such different habitats. Driven by the knowledge that amphibians produce several metabolites mainly as an essential self-defense strategy, recently drug discovery dip into them. Many peptides with anticancer activities have been found in amphibian skin or oocyte cells and eggs (Table 3) [105].

A sialic acid-binding lectin (SBLc), better known as leczyme, is a multifunctional protein isolated from the oocytes of *Rana catesbeiana* and characterized by a dual nature: a lectin and RNAse activity. This nature is the key to its very interesting antitumor properties (Figure 3). This lectin is unique and not homologous to any other known protein. It is a single subunit of 111 residues of amino acids and does not contain any covalently bound carbohydrate. The amino terminus is a pyroglutamyl residue and all the half-cystinyl residues are in the form of oxidized disulfide bridges [129]. SBLs lectin-glycan binding has not been completely elucidated. Nitta et al. [130] showed that Lectin hemagglutination activity was inhibited blocking amino groups and in particular t-amino ones, but not tyrosine residues, letting presume that those residues are responsible for the binding with the sugar.

The role of the lectin part of SBLc is to select cancer cells through the binding of specific sialic acid carbohydrates residues; then, specific receptors allow it to penetrate the cells, where the RNAse activity promotes the cleavage of RNA. Different structural changes on SBLc demonstrated that both intracellular incorporation and RNAse activity play a pivotal role for its anticancer activity. Indeed, increased incorporation with moderately increased RNAse activity is more effective than the increase in total RNAse activity [106,109]. For example, a less efficient internalization system does not affect or affects to a lesser extent the cytotoxicity of SBLc on estrogen negative cell lines, such as SK-BR3 [111,116]. On the other hand, a significant decrease in RNAse activity compromises SBLc’s antitumor activity [111]. In other words, as Aristotle would suggest, *in medio stat virtus*.

When inside the cell, SBLc activates the intrinsic apoptotic pathway on more than 30 tumor cell lines representing leukemia [107,108,109,118,120,131,132], Burkitt’s lymphoma [108,109], cervical [133], hepatocellular [120,121,133], and estrogen-positive and -negative breast carcinomas [113,114,116,117,120], glioblastoma [119], and malignant mesothelioma [110,113] (Table 3).

Just as in a jigsaw puzzle, many studies have tried to elucidate the proper molecular mechanism of action of SBLc. The pieces of the puzzles are a tumor-specific mechanism of action, the activation of the intrinsic apoptotic pathway with the involvement of heat shock protein 70 (HSP70) and the phosphorylation of p38 MAPK [108,110,116,120,134,135].

On MCF-7 estrogen receptor (ER)-positive breast cancer cells and H28 malignant mesothelioma cells, SBLc induced Bcl-2-mediated cell death [116]. On Jurkat cells, it promoted endoplasmic reticulum stress that may or may not be involved in its antitumor activity [134]. On p388 mouse lymphoma cells, HSP70 was linked to its ability to cause apoptosis. Even though the mechanism through which HSP70 assists SBLc cytotoxicity is not clear, it does not involve the binding between SBLc and cell membrane [109,135]. Finally, even if mitochondrial disruption is often part of SBLc mechanism of action, this lectin promoted a caspase-independent modality of cell death in SK-Hep1 hepatocellular carcinoma cells [120].

The activation of p38 MAPK pathway by SBLc has been recorded in leukemia, mesothelioma, and several breast cancer cells [108,111]. In these cell lines, the phosphorylation of p38 MAPK plays a key role in the pro-apoptotic activity of SBLc since the ablation of this pathway significantly decreases SBLc-mediated activation of caspase-3 and caspase-7 and cleavage of PARP and reduced cell death [111,114]. 

Special reference needs to be made to understand SBLc activity on breast cancer. The expression of ER, progesterone receptor (PR) and the human epidermal growth factor receptor 2 (HER2) represents the groundwork for the classification of this type of cancer and is useful for prognostic prediction and therapy selection. Usually, the worst prognosis refers to triple negative cancer (ER, PR and HER negative) and to those tumors overexpressing HER2 together with the epidermal growth factor (EGFR) [113]. Different studies about SBLc antitumor ability showed different results on the same tumor types. This is probably due to the different experimental procedures used in the studies, such as the presence of hormones, growth factor or fetal bovine serum in the medium. All studies agree on the fact that SBLc is very active on ER-positive cells, such as MCF-7 and ZR-75-1 [111,114,116], while contradictory data are published on ER-negative cell lines. For Tseng and colleagues [116], ERs are essential for cell binding of SBLc and thus its internalization. Accordingly, they reported a very mild activity of SBLc on MDA-MB-231 and ZR-75-30 ER-negative cell lines. On the contrary, Tatsuta et al. [114] and Kariya et al. [111] showed that both ER-negative (MDA-MB-231) and HER2-positive are sensitive to SBLc. Furthermore, Tatsuta et al. [114] showed a comprehensive screening of SBCc cytotoxic on several breast cancer cell lines with different phenotypes, such as ER-positive, PR-positive, HER2-positive, and triple-negative. As above mentioned, they demonstrated the involvement of p38 MAPK signal in its cytotoxic activity together with the modulation of the anti-apoptotic proteins of the Bcl-2 family on all cell lines. On MCF-7 and ZR-75-1, SBLc downregulated ERα, PR and HER2, while no modulation of EGFR was observed. This result is particularly interesting since hormonal therapies, and in particular receptor antagonism, represent efficient and usually well-tolerated treatments for breast cancer [136]. The cause of ER and PR downregulation could be linked to the activation of kinases, such as p38, drove by the ubiquitin proteasome system. Since SBLc promotes the phosphorylation of this kinase, Tatsua et al. [114] suggest that SBLc-promoted p38 phosphorylation can be linked to the destruction of hormone receptors. Likewise, the ubiquitin proteasome system may be involved in HER2 downregulation [114]. The hypothesis is based on the observation that SBLc modulates HSP70 [109,135]. Indeed, HSP70, together with HSP90, is responsible for the delivery of proteins, such as HER, to the ubiquitin-proteasome system in order to be degraded [114].

To avoid inflammation and promote dying-cell clearance, many types of specialized cells can phagocyte them. Cancer cells are not the first kind of professional phagocytes that come to mind; however, in rare situations, they are able to do that. It was already demonstrated that MCF-7 cells can play that role [137]. SBLc itself stimulated morphologic and phenotypic changes that allow viable MCF-7 cells phagocyting SBLc-triggered dying cells [115], promoting a very efficient auto-clearance. This unusual behavior represents an added value to the already promising anticancer activity of SBLc.

For many types of cancer, combination therapy is the best way to increase therapeutic efficacy, avoid drug resistance, and lessen the cytotoxicity of each single compound used in the therapy [138]. For example, malignant mesothelioma is treated with a combination of pemetrexed and cisplatin. SBLc alone was effective on cisplatin- and pemetrexate-resistant MESO-1 and MESO-4 cells (Table 3). Moreover, it synergistically increased the antitumor activity of pemetrexate in H28 [138] and MSTO cells [113] (Table 3) with an even stronger effect than that of pemetrexed plus cisplatin. The association of pemetrexed plus SBLs was particularly effective, because the first one blocks the proliferation of cells, which will be then easily killed thanks to the cytotoxic effect of the second one [113]. On the contrary, antagonistic effects were recorded when cisplatin was used together with SBLc. The antagonism is probably due to the opposite modulation of the tumor suppressor p21 by the two compounds. If cisplatin sharply increases p21 expression, SBLc antagonizes it, resulting in a lack of synergy [112]. In the case of SBLc plus TNF-related apoptosis-inducing ligand, a synergistic effect on H28 cells has been recorded, probably caused by an enhancement in the cleavage of the proapoptotic protein Bid that reinforces caspase activation [110]. SBLc also synergizes the antitumor cytotoxicity of interferon-γ (IFN-γ) in a cell-type-dependent way on different cell lines. The most marked effect occurred on MCF-7 cells. For the other tested cell lines, the degree of cell differentiation seems to play a role in this synergism: less degree of cell differentiation, more synergistic activity of SBLc plus IFN-γ. Then, an increasing trend of potency accompanied the well-differentiated HepG2 cells, the intermediate-differentiated J5 cells, and the poorly differentiated SK-hep1 cells, respectively. No evidence of synergism has been reported on HL-60 cells [120]. Considering that IFN-γ promotes differentiation of HL-60, this explanation perfectly fits with the abovementioned remarks [120]. Even if the vast majority of studies ascribe the SBLc’s selectivity to its lectin nature (i.e., to its ability to bind tumor cells because of their membrane glycosylation pattern), a correlation with the degree of differentiation of target cells and the ability of SBLc to recognize them is probably involved here as well. Indeed, the antitumor activity of SBLc decreases with the increase of differentiation of both hepatoma [121] and leukemia cells [118]. SBLc does not affect any of the several non-transformed cell lines tested that are all characterized by a high degree of differentiation: fibroblasts [109,120,121,133,139], epidermal melanocytes [109], keratinocytes [109], and mesothelial cells [110].

The effective and safe profile of SBLc reported above was also observed in animal models. Already in 1994, Nitta et al. [107] demonstrated the ability of SBLc to counteract sarcoma 180 cells inoculated on mice (Table 3) with doses at least 150 times lower than the lethal ones. More, SBLc suppressed mesothelioma (H2452 or MSTO cells) growth in xenografted nude mice, without causing toxicity nor body weight changes [113]. For the record, pemetrexed, which is one of the drugs of choice currently used for the cure of mesothelioma, failed to eradicate murine MSTO-xenografted tumors on the same experimental conditions [113]. In addition, SBLc effectively inhibited the growth of glioblastoma tumors in nude mice once again without causing any side effect [119].

### 5.2. Fish

The economic value of fish is not entirely due to the food industry. On the contrary, many pieces of research are focusing on these animals since they produce many bioactive compounds, such as the human calcitonin from salmon used for the treatment of postmenopausal osteoporosis [140]. As for amphibians, fishes produce many types of lectin that have been isolated from eggs, skin mucus, plasma, and serum [141,142,143,144]. The biologic function of fish lectin is not crystal clear, even if evidence shows some implication on fertilization and morphogenesis and a defense activity versus microorganisms [145], whether for some of them, an antitumor activity has been proven (Table 3).

*Aristichthys nobilis*, commonly known as bighead carp, belongs to the *Cyprinidae*, from the same family as zebrafish. It produces a lectin, GANL, which causes tumor-type-dependent cell death. GANL is a homo-multimeric glycoprotein that does not require Ca^2+^ ions to perform its functions. The carbohydrate content is approximately 13.4%, while the protein part is enriched in Asp, Glu, Leu, Val, and Lys. These amino acids organize themselves to form α-helices, unordered structures, β-turns, and β-sheets [124,146]. GANL blocked HeLa, SKOV3 (ovarian cancer cells), and HepG2 proliferation in a concentration-dependent manner while no effect was recorded on SMMC-7721 carcinoma cells and BGC803 gastric cancer cells (Table 3). Yao et al. [124] suggested that this behavior could be a consequence of the different glycosylation pattern that characterizes the different cells since it is not known whether the binding site of the lectin is the same for all cell lines and how each cell line expresses it [124].

The α-galoctoside-binding lectins isolated from the eggs of the catfish *Silurus asotus* (SAL) and the chum salmon eggs (CSEL) recognize specifically Gb3 sugar chain. Thanks to Gb3, SAL and CSEL bind cells, are internalized, and exhibit antitumor activity. Accordingly, SAL is not active on the Gb3-devoid K562 cells, while several studies showed its antitumor activity on Burkitt’s lymphoma cells [127,128,147] (Table 3). Similarly, CSEL promotes apoptosis on Gb3-positive Caco-2 cells, but it does not exert any effect on Gb3-negative DLD-1 colorectal adenocarcinoma cells [125]. On Raji cells, SAL induces the typic phenotypic changes of apoptosis, such as phosphatidylserine exposition and cell shrinkage [127], but its antitumor activity is more probably linked to its ability to inhibit the cell cycle [128]. A family of kinase proteins regulates rate and progression of the cell cycle. These enzymes consist of a regulating subunit (which takes the name of cyclin) and a catalytic subunit that takes the name of kinase-cyclin-dependent proteins (CDK). CDKs are inactive in the absence of the cyclin subunit and become active only when cyclins bind to the catalytic subunit. CDKs act during particular moments of the cell cycle *via* phosphorylation activity, stimulating or inactivating, the specific proteins that modulate cell-cycle advancement. For their part, synthesis and deactivation of cyclins are strictly regulated within the different phases of the cell cycle in order to allow this process running correctly [148]. After binding Gb3, SAL activated the GTPase Raf and the two kinases MEK and ERK. In turn, they promoted the synthesis of p21 [128], which physiological function is to regulate cell-cycle progression at G1 and S phase [149]. Moreover, SAL increased the expression of CDK4, c-myc, and cyclin D3. The overall effect is a G0-G1 cell-cycle arrest [128].

The Chinook salmon, also known as king salmon for its massive dimensions, is the source of another antitumor lectin, called rhamnose-binding chinook salmon roe lectin (CSRL). CSRL was reported to inhibit MCF-7 and HepG2 cell proliferation. HepG2 resulted in being more sensitive to CSRL than MCF-7 with a double potency (Table 3). As for all lectins mentioned in this review, CSRL as well did not exert any toxic effect on WRL68 non-transformed liver cell line at the same concentration used for MCF-7 and HepG2 cells [126] (Table 3).

Protein arginine methyltransferases (AMTs) are proteins involved in several processes, such as cell proliferation, cell differentiation, and also tumorigenesis. They catalyze the methylation of specific arginines on several nuclear and cytoplasmic substrates. AMT-5’ main targets are specific histones, and the result of their methylation is the silencing of different genes [150]. Recently, the transcription factor E2F-1 has been identified as a specific substrate for symmetric methylation, behaving as AMTs downstream element. E2F-1 modulates both apoptosis and cell-cycle progression, depending on which member of AMT family activates it. In particular, E2F-1 methylation by AMT-5 promotes cell proliferation, while if AMT-1 activates it, apoptosis is triggered [150]. p53 and p73 (the proline 73 polymorphic variant of p53) represent the links between EF2-1 and apoptosis [151,152]. Two different lectins are able to trigger this pathway and promote apoptosis in Hep3b cells. *Dicentrarchus labrax* fucose-binding lectin (DlFBL) and *Anguilla japonica* lectin 1 (AJL1) were harbored in a replication-defective adenovirus, generating respectively DIFBL-FLAG and AJL1-FLAG. DIFBL-FLAG and AJL1-FLAG elicited cytotoxicity in several human liver and lung cancer cell lines [61,123] (Table 3). DIFBL is a Ca^2+^-independent non-glycosylated lectin. It is a dimeric protein composed by two protein fractions, and, if they are separated, only one of them has lectin activity, while in physiological conditions they are stabilized by disulfide bonds [153]. Additionally, AJL1 has beta-galactoside specific activity in a Ca^2+^-independent manner. This lectin is composed of 142 amino acid residues having no half-cysteinyl residues, and exists in the form of homodimer without any covalent bonds [144]. On Hep3B cells, both lectins promoted apoptosis through the modulation of the AMT-5 pathway. In particular, the involvement of this enzyme provoked the downregulation of E2F-1 that in turn called Bcl-2 apoptotic protein family into play. Indeed, exogenous DIFBL and AJL1 downregulated Bcl-2 and XIAP proteins [61,123]. AJL1 also downregulated p38 MAPK and ERK protein levels, but it is not clear if this event is linked to its ability to promote apoptosis [123]. 

Despite the remarkable biological activity of lectins transported by viral vectors, some of them struggle to significantly infect cells, due to the lack of specific receptors such as coxsackie-adenovirus receptor (CAR). To facilitate this process, it is possible to target cell membrane receptors with other oncolytic adenoviruses that conveys a CAR-ligand-expression cassette in its genome. In this way, after infection and replication, the expression of CAR ligand will help the following adenovirus infection, triggering a positive feedback mechanism [57]. For this reason, a CAR-DIFBL was built and used to infect K562/adr doxorubicin-resistant leukemia and U87MG glioblastoma cells. CAR-DIFBL was able to favor cell infection by DIFBL-FLAG and also to synergize its cytotoxic activity [57] (Table 3). However, attention must be paid because another CAR-lectin, CAR-HddSBL, did help DIFBL-FLAG to infect glioblastoma cells, but it counteracted its cytotoxicity probably upregulating E2F-1 transcription levels [57].

## 6. Conclusions

The role of marine and freshwater lectins as anticancer agents has been discussed in this review and sustained by the reports summarized here. Marine and freshwater lectins exploit their antitumor action triggering apoptosis and other forms of programmed cell death, inhibiting cell cycle and blocking neoangiogenesis (Figure 4). 

Some of them are also able to improve the antineoplastic activity of standard antitumor drugs. Moreover, all the lectins described in this review disclosed distinctive features against different types of tumors and, in the majority of the times, their ability to distinguish between normal and transformed cells. Of note, all lectins tested on animals showed optimal antitumor activity matched by negligible toxicity. Thus, selectivity and cytotoxicity are the two main characteristics that promote lectins to the rank of ideal antitumor agents. Indeed, toxicity of lectins of plant origin has been recorded only after direct ingestion of significant amount of specific lectins, such as phytohemagglutinin, which is particularly enriched in raw kidney beans. Gastric symptoms, such as nausea, vomiting, and diarrhea, are the most common ones after oral acute exposure, while immune system impairment, lung hypertrophy, and other systemic effects are very rare. On the contrary, no significant adverse reactions have been documented if lectins were administered as drug through alternative ways [154]. 

Another aspect that this review unearthed is that marine lectins are often tumor-type specific. This is not surprising because, going back to tumor heterogenicity, the glycosylation pattern is not the same for all cancer cells. This evidence does not represent a limiting factor for lectins. Indeed, marine biodiversity comes to our aid, and if one tumor is not sensitive to one specific lectin, most likely it would be to another one. 

So far, the most concerning issue about exploiting marine compounds for a pharmacological purpose is that lectins, as all marine metabolites, are not produced in vast quantities. To obtain a reasonable amount of them, a large number of organisms has to be harvested. Thus, a significant limiting factor that stands between lectins and their exploitation as therapeutic agents is their availability. In other words, one of the hardest challenges about lectins is moving from laboratory scale to bulk production. However, despite the peculiarity of lectins that does not allow an easy identification of the aminoacidic bone structure and a cheap synthesis de novo, aquaculture or mariculture could bridge the gap. Additionally, the easiest and most efficient way to produce lectins is to exploit heterologous systems, such as bacteria and yeast. Indeed, different synthesis strategies exploit genetic engineering to transfer the genes encoding the lectin of interest to microorganisms, which can be grown in vast quantities, or to virus vectors in order to allow the synthesis of the lectins directly inside tumor cells [26,155,156]. As presented above, the latter approach has been exploited adequately for several lectins and with considerable results.

Despite the outstanding potential of marine lectins demonstrated by preclinical studies, so far, no clinical trials are translating this knowledge to cancer patients. In this context, terrestrial lectins are paving the way in that direction. For instance, mistletoe lectin extract has been subjected to many phase-I clinical trials, with promising, although not conclusive, results [157]. However, bearing in mind that the drug development process can last up to decades, preclinical studies represent the foundation to draw clinical trials. 

Taking all this data together, it follows that aquatic organisms represent an important source of lectins and that those proteins represent a bright future for anticancer research. Certainly, further studies are needed to understand the actual antitumor potential of lectins on humans full. To reach this aim, clinical studies are essential to progress cancer research in employing lectins for antineoplastic care, but the premises for outstanding outcomes are all there.

## Figures and Tables

**Figure 1 marinedrugs-18-00011-f001:**
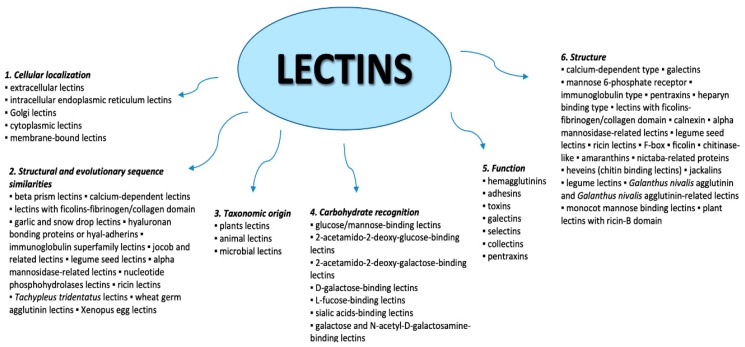
Classification of lectins.

**Figure 2 marinedrugs-18-00011-f002:**
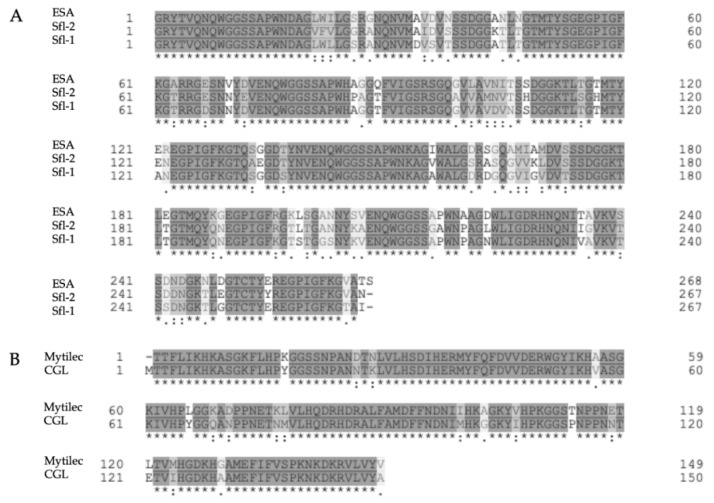
Comparison of amino acid sequence of ESA, Sfl-1 and Sfl-2 (**A**) and Mytilec and CGL (**B**). Data were analyzed for protein sequence similarity using BLAST technology, through the data base UniPROT [45]. “*” identical residues; “:” conserved substitution; “.” Semi-conserved substitution; “_” gaps.

**Figure 3 marinedrugs-18-00011-f003:**
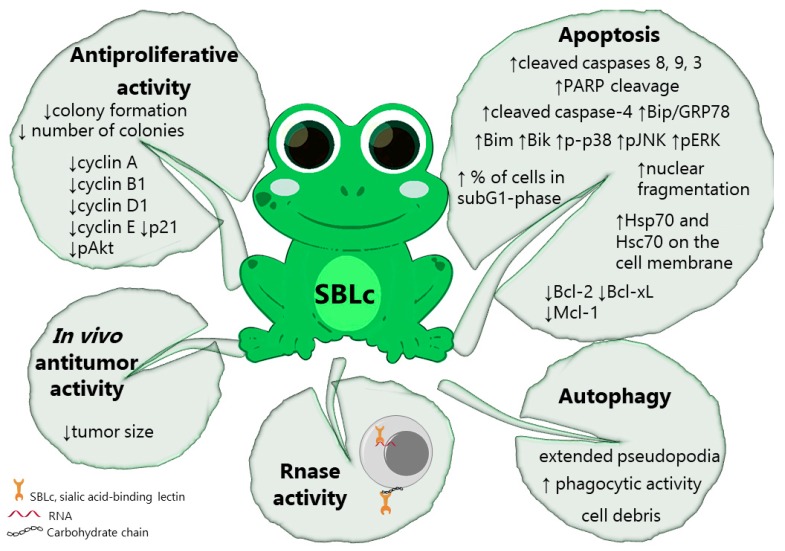
Anticancer mechanisms of SBLc, isolated from the oocytes of *Rana catesbeiana*.

**Figure 4 marinedrugs-18-00011-f004:**
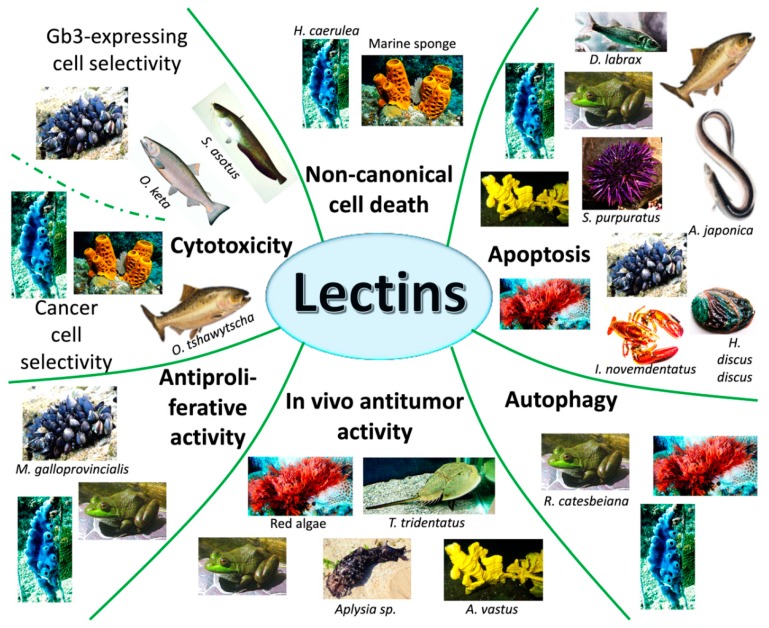
Lectins isolated form marine and freshwater organisms induce anticancer effects in cell cultures and animal models.

**Table 1 marinedrugs-18-00011-t001:** Antineoplastic effects of lectins from algae based on in vitro and in vivo studies.

Origin	Lectin	Recognition Glycans	Cell Lines	In Vivo Model	Treatment Times and Doses	Cellular and Molecular Target	Ref.
*Acanthophora spicifera*	Crude lectins fraction		MCF-7		24 h (100 µg/mL)	% of cell growth inhibition in *A. spicifera*: 1.78 µg/mL (MCF-7); 4.27 µg/mL (HeLa)	[35]
*Acrocystis nana*			HeLa			% of cell growth inhibition in *A. nana*: 9.10 µg/mL (MCF-7); 47.68 (HeLa)	
*Eucheuma serra*	*Eucheuma serra* agglutinin (ESA)	Mannose	Colon26	Colon-26 cells injected in BALB/c mice	48 h (0–1000 µg/mL)	↓viability at concentrations > 8 µg/mL	[36]
					48 h (50 µg/mL)	% of AnnexinV^+^/propidium iodide^−^: 31.4%	
						↑caspase-3 activity	
					400 mg /200 mL PBS every 3 days up to 15 days (intravenously injection)	↓tumor volume	
						TUNEL-positive cells in tumor	
*Eucheuma serra*	*Eucheuma serra* agglutinin (ESA)	Mannose	OST		24 h (10–50 µg/mL)	Cell viability (50 µg/mL): 41.7 ± 12.3% (LM8); 54.7 ± 11.4% (OST)	[37]
			LM8		3–4 h (50 µg/mL)	AnnexinV^+^/propidium iodide^−^ (3 h): 68.2% (LM8); 74.8% (OST) AnnexinV^+^/propidium iodide^−^ (24 h): 24.1% (OST); 68.8% (LM8)	

					16 h (50 µg/mL)	OST: ↑caspase-3 activity (2.3-fold increase)	
	PEGylated vesicles with immobilized ESA (EPV)				24 h (1–5 µg/mL of ESA delivered by EPV)	Cell viability (1 µg/mL): ∼50% (OST)	
*Eucheuma serra*	*Eucheuma serra* agglutinin (ESA)	Mannose	Colo201		72 h (0.05–150 µg/mL)	↓Viability at concentrations > 1.2 µg/mL (cancer cells)	[38]
			HeLa			No cytotoxicity at 10 µg/mL (MCF10-2A)	
			MCF-7				
					24 h (64 µg/mL)	DNA degradation (Colo201)	
			MCF10-2A		16 h (10.8 µg/mL)	↑caspase-3 activity (Colo201)	
*Eucheuma serra*	Span 80 vesicles containing immobilized ESA (EV) Span 80 vesicles containing DSPE-PEG2000 and immobilized ESA (EPV) Span 80 vesicles containing DSPE-PEG2000, immobilized ESA and entrapped ESA (EEPV)	Mannose	Colo201		0–24 h (54 µg/mL of ESA delivered by EV)	Cell viability (24 h): 17.2% (Colo201); no effect (MCF10-2A)	[39]
			MCF-7				
			Colon26		8 h (54 µg/mL of ESA delivered by EV)	DNA fragmentation in Colo201 and MCF-7	
			MCF10-2A				
				Colo201 cells transplanted in Balb/c-nu/nu mice	EPVs (containing 2.0 µg/mL of ESA) or EEPVs (containing 2.5 µg/mL of ESA) (0.01 mL/g b.w.) injected every 3 days up to 15 days	↓tumor volume (9th day): 51.1% (EEPV); 58.0% (EPV)	
					3 days after EPVs (0.01 mL/g b. w.) injection	TUNEL-positive cells around the blood vessels	
*Gloiocladia repens*	Crude lectins fraction		MCF-7		24 h (100 µg/mL)	% of cell growth inhibition in G. repens: 14.19 µg/mL (HeLa); 28.54 µg/mL (MCF-7)	[35]
*Helminthora divaricata*			HeLa			% of cell growth inhibition in H. divaricata: HeLa cells: 3.63 µg/mL (HeLa); 12.25 µg/mL (MCF-7)	
*Microcystis viridis*	Recombinant *Microcystis viridis* lectin (R-MVL)	Mannose	HT-29		72 h (2–64 µg/mL)	IC50 ^1^: 40.20 µg/mL (SCG-7904); 42.67 µg/mL (HepG2); 49.87 µg/mL (HT-29); 53.40 µg/mL (SKOV3)	[40]
			HepG2				
			SKOV3				
			SCG-7904				
*Nitophylium punctatum*	Crude lectins fraction		MCF-7		24 h (100 µg/mL)	% of cell growth inhibition: 2.97 (HeLa); 15.53 (MCF-7)	[35]
			HeLa				
*Solieria filiformis*	*Solieria filiformis* lectin (SfL) (mixture of isoforms 1 and 2)	Mannopentose	MCF-7		24 h (0–500 µg/mL)		[41]
			HDA				
					24 h (125 µg/mL)	AnnexinV^+^/propidium iodide^−^: 25.07%; AnnexinV^+^/propidium iodide^+^: 35.16%	
						↓*Bcl-2*; ↑*Bax*, ↑*caspase-3*, ↑caspase-*8*, ↑caspase-*9*	
*Ulva pertusa*	Adenovirus-*Ulva pertusa* lectin 1 (Ad-UPL1)	*N*-acetyl d-glucosamine	Huh7		48 h (100 MOI ^2^ Ad-UPL1 ± 10 µM U0126 ^3^)	Cell viability: ∼50% (Ad-UPL1 + U0126, Huh7); ∼90% (Ad-UPL1, BEL-7404 or Huh7)	[42]
			BEL-7404		48 h (50–100 MOI)	Huh-7: ↑pERK1/2, p-p38; ↓Akt	
						BEL-7404: ↑pERK1/2; ↓Akt	
						Huh-7: ↓Beclin-1; ↑LC3-II	
						BEL-7404: ↑Beclin-1; ↓LC3-II	

^1^ Half maximal inhibitory concentration; ^2^ multiplicity of infections; ^3^ MEK 1/2 inhibitor.

**Table 2 marinedrugs-18-00011-t002:** Antineoplastic effects of lectins from invertebrates based on in vitro and in vivo studies.

Origin	Lectin	Recognition Glycans	Cell Lines	In vivo Model	Treatment Times and Doses	Cellular and Molecular Target	Ref.
**Arthropoda and Mollusca**							
*Aplysia kurodai* eggs	*Aplysia kurodai* egg lectin (AKL)	Galactose		Brine shrimp nauplii	24 h (2–32 µg/mL)	Mortality (32 µg/mL): 33.33% (PnL); 63.33% (AKL)	[52]
*Crenomytilus grayanus*	*Crenomytilus grayanus* lectin (CGL)	Galactose-lactosylceramide	MCF-7		24 h (50–200 µg/mL)	Cell viability: 33% at 200 µg/mL	[53]
*Crenomytilus grayanus*	*Crenomytilus grayanus* lectin (CGL)	Galactose-lactosylceramide	Raji		48 h (0–100 µg/mL)	IC50: 6.81 ± 0.83 µg/mL (Raji); unaffected (K562)	[54]
			K562		24 h (10 µg/mL CGL + 100 mM glucose, lactose, melibiose, raffinose or galactose)	Cell viability (Raji): ∼55% (CGL + Lactose); ∼70% (CGL + Glucose); ∼100% (CGL + Melbiose, raffinose or galactose)	
					24 h (2.5–20 µg/mL)	↑% of cells in sub-G1 and G2/M phases and ↓G1 and S phases	
					24 h (2.5–20 µg/mL)	AnnexinV^+^/propidium iodide (5 µg/mL): ∼70%	
					24 h (2.5–10 µg/mL)	↑ caspase-9, caspase-3 and PARP cleavage	
*Haliotis discus discus*	*Oncolytic vaccinia* virus (oncoVV)-*Haliotis discus discus* sialic acid-binding lectin (HddSBL)	Sialic acid	C6	C6 tumor-bearing athymic BALB/c nude mice	60 days (10^7^ pfu ^1^/twice)	Mice survival: oncoVV-HddSBL > onco-VV	[55]
					15 days (10^7^ pfu)	IL-2 secretion: oncoVV-HddSBL < onco-VV	
					24 h (5 MOI)	mRNA IL-2: oncoVV-HddSBL < onco-VV	
						NF-kB and AP-1 activity: oncoVV-HddSBL > onco-VV	
						IFIT2, IFIT3; DDX58: oncoVV-HddSBL < onco-VV	
					2–36 h (2 MOI)	OncoVV-HddSBL replication > onco-VV	
*Haliotis discus discus*	Adenovirus (Ad.FLAG)-*Haliotis discus discus* sialic acid binding lectin (Ad.FLAG-HddSBL)	Sialic acid	Hep3B		96 h (1–20 MOI)	Cell viability (20 MOI): ∼40% (Hep3B); ∼50% (A549); ∼60% (H1299); ∼80% (SW400)	[56]
			A549		48 h (20 MOI)	AnnexinV^+^/propidium iodide (Hep3B): 19.8% (Ad.FLAG-HddSBL) vs 4.79% (Ad.FLAG)	
			H1299			↓Bcl-2;	
			SW480				
*Haliotis discus discus*	*Haliotis discus discus* sialic acid binding lectin (HddSBL) + coxsackie-adenovirus receptor (sCAR-HddSBL)	Sialic acid	K562/adr		48 h (5–30 MOI Ad-EGFP + 10 µg/mL sCAR-HddSBL)	Viral infection and replication: 13% sCAR-HddSBL vs 3.19% Ad-EGFP (K562/ADR); 48.6% sCAR-HddSBL vs 23.1% Ad-EGFP (U87MG)	[57]
			U87MG		96 h (8.2 MOI Ad-DlFBL + 10.6–31.8 µg/mL sCAR-HddSBL)	Cell viability (U87MG): ∼40% (Ad-DlFBL); ∼90% (31.8 µg/mL Ad-DlFBL-sCAR-HddSBL)	
					48 h (8.2 MOI Ad-DlFBL + 31.8 µg/mL sCAR-HddSBL)	U87MG: AnnexinV^+^/propidium iodide^-^ 10.2% (Ad-DlFBL-sCAR-DlFBL) vs 7.91% (Ad-DlFBL)	
					48 h (8.2 MOI Ad-DlFBL + 31.8 µg/mL sCAR-HddSBL)	U87MG: ↑pERK (Ad-DlFBL-sCAR-HddSBL); ↑E2F1 (sCAR-HddSBL; Ad-DlFBL-sCAR-HddSBL)	
*Ibacus novemdentatus*	*N*-acetyl sugar-binding lectin (iNol)	*N*-acetylated glycan	MCF-7		48 h (0–100 µg/mL)	IC_50_: 12.5 µg/mL (Caco2); 25 µg/mL (HeLa); 50 µg/mL (MCF-7); 100 µg/mL (TD47D)	[58]
			T47D		24 h (0–100 µg/mL)	HeLa: ↑caspase-9	
			HeLa		12–48h (100 µg/mL)	HeLa: ↑caspase-3 activity	
			Caco2		24 h (100 µg/mL)	HeLa: ↑DNA degradation; chromatin condensation	
*Mytilus galloprovincialis*	𝜶-d-galactose-binding lectin (MytiLec)	Galactose-lactosylceramide	Raji		24 h (0-50 µg/mL)	Cell viability (50 µg/mL): ∼40% (Raji); ∼100% (K562)	[59]
			K562		24 h (20 µg/mL MytiLec + 100 mM Sucrose, Lactose or Melbiose)	Cell viability (Raji): ∼40% (MytiLec + Sucrose or Lactose); ∼100% (MytiLec + Melbiose)	
*Mytilus galloprovincialis*	𝜶-d-galactose-binding lectin (MytiLec)	Galactose-lactosylceramide	Ramos		24 h (0.5–50 µg/mL)	Cell viability (50 µg/mL): ∼45% (Raji); ∼100% (K562)	[60]
			K562		12–24 h (0.5–50 µg/mL)	↑pMEK, pERK and p21; ↓CDK6, ↓cyclinD3	
					12–24 h (20 µg/mL)	↑pJNK, ↑pp38, ↑pERK	
					12–24 h (20 µg/mL)	↑caspase-9, ↑caspase-3, ↑TNF𝜶	
					12–24 h (20 µg/mL + 10 µM U0126- pMEK inhibitor)	↑caspase-9, ↑caspase-3	
*Perinereis nuntia*	*Perinereis nuntia* lectin (PnL)	Galactose		Brine shrimp nauplii	24 h (2–32 µg/mL)	Mortality (32 µg/mL): 33.33%	[52]
*Strongylocentrotus purpuratus*	Adenovirus FLAG (Ad.FLAG)-*Strongylocentrotus purpuratus* rhamnose-binding lectin (SpRBL)	Rhamnose	Hep3B		48–96 h (1–20 MOI)	Cell viability (20MOI; 96 h; Ad.FLAG-SpRBL): ∼20% (Hep3B); ∼30% (BEL-7404, A549); ∼40% (SW480)	[61]
			BEL-7404		48 h (20 MOI)	Hep3B: Annexin V^+^/propidium iodide^-^: 25.4% (Ad.FLAG-SpRBL) vs 1.35% (Ad.FLAG)	
			A549			Hep3B: = cleaved PARP; ↓Bcl-2, ↓XIAP	
			SW480		48 h (20 MOI)	Hep3B: ↓E2F-1	
*Tachypleus tridentatus*	*Oncolytic vaccinia* virus (oncoVV)-*Tachypleus tridentatus* Lectin (TTL)	Rhamnose	MHCC97-H	MHCC97-H tumor-bearing athymic BALB/c nude mice	44 days (10^7^ pfu/twice)	↓tumor volume	[55]
					36 h (5 MOI)	OncoVV-TTL replication > onco VV	
			BEL-7404		24 h (5 MOI)	↑pERK (onco VV = oncoVV-TTL)	
						MAVS, IFI16, IFNβ: ↑ (oncoVV); = (onco VV-TTL)	
					(5 MOI ± U0126)	U0126 ↓ oncoVV-TTL replication	
**Chordata**							
*Didemnum ternatanum*	*Didemnum ternatanum* lectin (DTL)	N-acetyl-D-glucosamine	HeLa in adhesion plates		72 h (2.5 µg/mL)	Cell proliferation: ∼50%	[62]
			HeLa in soft agar		2 weeks (2.5 µg/mL)	Colony formation: 7 ± 1 (control); 25 ± 2 (DTL in agar); 60 ± 4 (DTL in plates and in agar)	
**Porifera**							
*Aphrocallistes vastus*	*Oncolytic vaccinia* virus (oncoVV)-*Aphrocallistes vastus* lectin (AVL) Adenovirus (Ad)-*Aphrocallistes vastus* lectin (AVL)	Galactose	HCT116		48–72 h (10–20 MOI Ad-AVL)	Cell Viability (20 MOI Ad-AVL; 72h): ∼40% (HCT116, U251); ∼50% (HT-29, MHCC97-H, BEL-7404)	[63]
			U251	BEL-7404 or HCT116 tumor-bearing athymic BALB/c nude mice	48-72 h (1-10 MOI Ad-AVL)	Cell Viability (5 MOI oncoVV-AVL; 72h): ∼20% (HCT116); ∼40% (U87, 4T1-LUC, BEL-7404)	
			BEL-7404		2–36 h (5 MOI)	OncoVV-AVL replication > onco VV	
			MHCC97-H		24 h (2 MOI)	AnnexinV^+^/propidium iodide (HCT116): 6.49% (oncoVV-AVL) vs 1.26% (oncoVV)	
			HT-29		24 h (2 MOI) on HCT116	MDA5: no effect	
			4T1-LUC			↓caspase-3 (oncoVV-AVL); ↓caspase-8, ↓Bax (oncoVV, oncoVV-AVL)	
			U87			↓NIK, pNF-𝜅B2 (oncoVV-AVL); ↑NIK (oncoVV)	
						↑pERK (onco VV = oncoVV-TTL)	
					24 h (5 MOI ± 10 µM U0126)	U0126 ↓ oncoVV-AVL replication	
					25 (BEL-7404) or 35 (HCT116) days (10^7^ pfu)	↓tumor volume	
*Cinachyrella apion*	Lactose-Binding Lectin (CaL)	Lactose	HeLa		24-48 h (0.5 - 10 µg/mL)	Cell Viability (10 µg/mL; 24h): ∼50% (HeLa); ∼60% (PC3); ∼75% (3T3)	[64]
			PC3		(10 - 20 µg/mL)	No cytotoxicity in peripheral blood cells	
			3T3		1 h (10 µg/mL)	No hemolysis in erythrocytes	
			Erythrocytes and peripheral blood cells		24 h (10 µg/mL)	Membrane blebbing and nuclear condensation	
					24 h (10 µg/mL ± 0.02 mM Z-VAD-FMK ^2^)	% of cells in S phase (HeLa): ∼40% (control); ∼ 50% (CaL + Z-VAD-FMK); 57.6% (CaL)	
						AnnexinV^+^/propidium iodide (HeLa): 3.84% (control); 15.5% (CaL + Z-VAD-FMK); 23.2% (CaL)	
					6–24 h (10 µg/mL)	HeLa: ↑Bax, ↑pNF-κB (105 kDa), ↑JNK; =Bcl2, =pAKT; ↓pNFkB (50 kDa)	
*Cliona varians*	*Cliona varians* lectin (CvL)	Galactose	Jurkat		72 h (1–150 µg/mL)	IC50: 70 µg/mL (K562); 100 µg/mL (Jurkat); no effect on lymphocytes	[65]
			K562		24 h (1–150 µg/mL)	No effect (B16, 786-O, PC3)	
			blood lymphocytes		72 h (70 µg/mL)	K562: ↑subG1 (28% CvL vs 14.1% control)	
			B16		72 h (50–70 µg/mL)	Apoptotic cells (K562; 70 µg/mL): 43% CvL vs 10% control	
			786-O		72 h (70 µg/mL)	K562: 25.3% (Annexin V-/propidium iodide+); 60.4% (Annexin V^+^/propidium iodide^+^);	
			PC3		72 h (50–70 µg/mL)	No increase in caspase-8, -9, and -3 activity	
					72 h (70 µg/mL)	Cathepsin B founded in cytoplasm and nucleus	
					2 h (5 µM E-64) + 72 h (50-80 µg/mL CvL)	Cell viability (80 µg/mL, K562): ∼30% (CvL); ∼100% (CvL + E-64)	
					72 h (50–70 µg/mL)	K562: ↑TNFR1, ↓NF-κB (p65 sub)	
						K562: ↑Bax, ↑Bcl-2	
						K562: ↑p21, ↓pRb	
*Haliclona caerulea*	Halilectin-3 (H3)	Mucin	MCF7		6-48 h (7.81–500 µg/mL)	Cell viability (250 µg/mL): 42% (MCF7); 75% (HDF); IC50: 100 μg/ml (MCF7)	[66]
			HDF		24 h (100 µg/mL)	↑% of cells in the G1 phase	
					24-48 h (100 µg/mL)	↑early apoptosis cells: 46% (24h); 55.4% (48h)	
					6-24 h (100 µg/mL)	24h: ↑caspase-3, ↑caspase-8, ↑caspase-9, ↑Bax, ↑TP53; ↓Bcl-2	
					8-24 h (100 µg/mL)	↑agglutination of MCF-7 cells; ↓cell adhesion	
					6 h (100 µg/mL)	↑LC3; ↓BECLIN-1	
						↑LC3II/LC3I	
						Autophagosoma vescicles	
*Haliclona cratera*	*Haliclona cratera* Lectin (HCL)	Galactose, *N*-Acetyl-d-galactosamine, Lactose	HeLa		48 h (0–40 µg/mL)	IC50: 9 µg/mL (HeLa); 11 µg/mL (FemX)	[67]
			FemX		72 h (0–15 µg/mL)	Lymphocytes: no toxicity	
			human T-lymphocytes		2 h (5 µg/mL PHA) + 72 h (0–15 µg/mL HCL)	Lymphocytes: 23% (PHA + HCL 15 µg/mL)	
*Halichondria okadai*	18 kDa Lectin (HOL-18)	*N*-acetylhexosamine	Jurkat		24 h (1–25 µg/mL)	Cell Viability (25 µg/mL): ∼30% (Jurkat); ∼60% (K562)	[68]
			K562		24 h (25 µg/mL HOL-18 ± 50 mM d-GlcNAc ^3^, d-GalNAc ^4^ or Mannose)	Cell Viability: ∼30% (Jurkat) - 50% (K562) (HOL-18 + Mannose); ∼80% (HOL-18 +d-GalNAc); ∼90% (HOL-18 + d-GlcNAc)	
*Halichondria okadai*	18 kDa Lectin (HOL-18)	*N*-acetylhexosamine	HeLa		48 h (6.25–100 µg/mL)	IC50: 40 µg/mL (HeLa); 52 µg/mL (MCF7); 63 µg/mL (T47D); no effect (Caco-2)	[69]
			MCF7		48 h (50 µg/mL HOL-18 + 20 mM Glucose, GlcNAc, Mannose, ManNAc ^5^)	Cell Viability: ∼45% (HOL-18 ± Glucose or Mannose); ∼75% (HOL-18 ± GlcNAC); ∼90% (HOL-18 ± ManNAC)	
			T47D		48 h (6.25–100 µg/mL)	HeLa: ↑pERK, ↑caspase-3	
			Caco2				

^1^ Plaque-forming unit; ^2^ carbobenzoxy-valyl-alanyl-aspartyl-[*O*-methyl]-fluoromethylketone; ^3^
*N*-acetyl d-glucosamine (d-GlcNAc); ^4^
*N*-acetyl d-galactosamine (d-GalNAc); ^5^
*N*-acetyl d-Mannosamine (d-ManNAc).

**Table 3 marinedrugs-18-00011-t003:** Antineoplastic effects of lectins from vertebrates based on in vitro and in vivo studies.

Origin		Lectin	Recognition Glycans	Cell Lines	In Vivo Model	Treatment Times and Doses	Cellular and Molecular Target	Ref.
**Amphibians**								
*Rana catesbeiana*		Sialic acid-binding lectin (SBLc)	Sialic acid	P388		48 h (0.1–5 µM)	IC50: 0.3 µM (EDC-ED SBLc); 1.0 µM (EDC-GM SBLc); 1.5 µM (EDC-TA SBLc; SBLc)	[106]
		EDC-TA SBLc						
		EDC-GM SBLc						
		EDC-ED SBLc						
*Rana catesbeiana*		Sialic acid-binding lectin (SBLc)	Sialic acid	P388		48 h (0.1–5 µM)	GI50 ^1^ (P388): 1.56 (SBLj); 6.25 µM (SBLc)	[107]
*Rana japonica*		Sialic acid-binding lectin (SBLj)		L1210			GI1002 (L1210): 1.56 µM (SBLc and -j)	
					Sarcoma 180-bearing ddY mice	A single SBLc injection (2.5–10 mg/kg)	IC50: 5 mg/kg (Sarcoma 180-bearing mice) after 45 days	
					MepII-bearing ddI mice		IC50: 10 mg/kg (MepII-bearing mice) after 45 days	
					Sarcoma 180-bearing ddY mice	Continuous SBLc injection (0.5–2 mg/kg) for 10 days	IC50: <0.5 mg/kg (Sarcoma 180-bearing mice) after 45 days	
					MepII-bearing ddI mice		IC50: 0.5 mg/kg (MepII-bearing mice) after 45 days	
*Rana catesbeiana*		Sialic acid-binding lectin (SBLc)	Sialic acid	Jurkat		48 h (2 µM)	44% of cells in sub-G1 phase	[108]
						1–48 h (2 µM)	↑cleaved caspase-8, -9, -3	
						3–48 h (2 µM)	↑cleaved caspase-4, ↑Bip/GRP78	
*Rana catesbeiana*		Sialic acid-binding lectin (SBLc)	Sialic acid	P388		24 h (3 µM)	Cell viability: ∼20% (P388); ∼30% (K562); ∼40% (HL60); ∼80% (MCF-7); ∼100% (Daudi; Raji; NHDF; NHEM; NHEK)	[109]
				K562				
				HL60		24 h (3, 20 µM)	No DNA fragmentation in Raji and NHDF cells	
				MCF-7			DNA fragmentation in P388 and K562 cells	
				Daudi		24 h (3 µM)	↑caspase-8, ↑caspase-3	
				Raji		1h (1 µM)	↑Hsp70 and Hsc70 on the cell membrane	
				NHDF				
				NHEM				
				NHEK				
*Rana catesbeiana*		Sialic acid-binding lectin (SBLc)	Sialic acid	H28		48 h (0.2–20 µM) of treatment and 12 days of posttreatment	Colony formation (5 µM): <5% (H28); 20% (MESO-4); <70% (MESO-1)	[110]
				MESO-1			Colony formation (5 µM): >90% (Met-5A)	
				MESO-4		24–72 h (5 µM)	Annexin V+ (72h): ∼5% (Met-5A); ∼15% (MESO-1 and -4); ∼50% (H28)	
				Met-5A		6–48 h (5 µM)	H28: ↑caspase-8, ↑caspase-9, ↑caspase-3	
							H28: ↑Bim, ↑Bik, ↑p-p38, ↑pJNK, ↑pERK	
						24 h (SBLc 5 µM ± TRAIL 2 ng/mL) on H28	↑cytotoxicity (∼30%) vs single treatment (∼70%)	
							↑ Annexin V+ (∼50%) cells vs single treatment (∼50%)	
							↑mitochondrial membrane depolarization vs single treatments	
							↑caspase-8, -9, -3 protein expression vs single treatment	
*Rana catesbeiana*		Sialic acid-binding lectin (SBLc)	Sialic acid	MCF-7		72 h (2 µM SBLc)	Cell viability: 25.5% (MDA-MB-231); 30.4% (MCF-7); 65.3% (SK-BR-3)	[111]
				SK-BR-3			↑p-p38	
				MDA-MB-231			↑ caspase-3/7 activity	
		SBLc mutant lacking RNase activity (H103A)				72 h (10 µM H103A)	Cell viability: 100% (MDA-MB-231)	
						72 h (2 µM H103A)	No effect on pp38, PARP expression	
							No effect on caspase-3/7	
*Rana catesbeiana*		Sialic acid-binding lectin (SBLc)	Sialic acid	H28		72 h (1–30 µM)	IC50: 0.46 µM (H28); 0.52 µM (H2452); 1.54 µM (MESO-4); 5.05 µM (MSTO); 5.51 µM (MESO-1); 52.22 (Met-5A)	[112]
				MESO-1		72 h (1 µM)	↑ % of cells in subG1-phase	
				MESO-4		72 h (1 µM)	↓cyclin A, ↓cyclin B1, ↓cyclin D1, ↓cyclin E, ↓p21, ↓pAkt	
				H2452		72 h (1 µM SBLc + 20 µM pemetrexed or 40 µM cisplatin)	CI3 (H28): 0.05 (SBLc + pemetrexed); 0.47 (SBLc + cisplatin)	
				Met-5A			Annexin V+/propidium iodide- (H28): no difference vs SBLc treatment	
							Caspase-3/7 activity (H28): no difference vs SBLc treatment	
							H28: ↑ % of cells in S- and subG1-phases (SBLc + pemetrexed); ↑ % of cells in S-, G2- and subG1-phases (SBLc + cisplatin)	
							H28: ↓cyclin B1, ↓p21, ↓pAkt (SBLc + pemetrexed or cisplatin)	
*Rana catesbeiana*		Sialic acid-binding lectin (SBLc)	Sialic acid	NCI-H2452	H2452 or MSTO injected in BALB/C nu/nu Slc	24–72 h (H2452: 1 µM; MSTO: 0.4 µM)	Annexin V+ (72 h): 16.13% (H2452); 40.05% (MSTO)	[113]
				MSTO-211H		6–72 h (H2452: 5 µM; MSTO 2 µM)	↑nuclear fragmentation (72h): ∼2.5-fold (H2452); ∼4-fold (MSTO)	
						1–72 h (H2452: 5 µM; MSTO 2 µM)	↑activity and expression of caspase-9, -8, -3	
						72 h [(1 pM–1 µM SBLc) + (0.8 nM–800 µM pemetrexed)]	CI (H2452) < 1 at all combinations (SBLc + pemetrexed or SBLc + cisplatin)	
						72 h [(1 pM–1 µM SBLc) + (0.1 nM–100 µM cisplatin)]	CI (MSTO) < 1 up to 1 µM SBLc + 1.5 µM pemetrexed or 10 µM cisplatin	
						Pemetrexed (100 ng/kg) on days 1–5 and 15–19	↓tumor size after 47 (H2452) days of treatment	
						SBLc (2.5 mg/kg) 2/week for 4 weeks	↓tumor size after 36 (H2452) or 29 (MSTO) days of treatment	
*Rana catesbeiana*		Sialic acid-binding lectin (SBLc)	Sialic acid	ZR-75-1		72 h (1–20 µM)	Cell viability (20 µM): 40% (MDA-MB-468); 45% (MCF-7); 46% (SK-BR-3); 51% (BT-474); 52% (MDA-MB-231); 69% (ZR-75-1); 85% (MCF10A)	[114]
				BT474		72 h (1–10 µM) + 7–28 days in drug-free medium	↓cell number (except for MCF10A)	
				MCF-7		72 h (10 µM)	↓ number of colonies (except for MCF10A)	
				SK-BR-3		72–96 h (10 µM)	chromatin condensation and nuclear collapse (except in MCF10A)	
				MDA-MB-231			↑cleaved caspase-9 and PARP cleavage (except in MCF10A)	
				MDA-MB-468		72 h (10 µM)	↑pp38, ↑pJNK (ZR-75-1); ↑pp38, ↓JNK (MCF-7)	
				MCF10A			↓Bcl-2, ↓Bcl-xL, ↓Mcl-1 (MCF-7); ↓Bcl-2, ↑Bcl-xL; Mcl-1 (ZR-75-1)	
							↓ER𝛂, ↓PgR, ↓HER2 (MCF-7); ↓ER𝛂, ↓HER2 (ZR-75-1)	
							↓ErbB family in each cancer cells	
						3–24 h (10 µM)	Triple negative cells: ↑pp38 (MDA-MB-231 and -468); ↓EGFR/HER1, ↓pAKT (only in MDA-MB-231 cells)	
*Rana catesbeiana*		Sialic acid-binding lectin (SBLc)	Sialic acid	ZR-75-1		120 h (20 µg/mL)	Cell survival (72h): ∼10% (MCF-7); ∼30% (ZR-75-1)	[115]
				MCF-7		72 h (20 µg/mL)	MCF-7, ZR-75-1: ↑caspase-3 activity	
							MCF-7: extended pseudopodia, increase in phagocytic activity, cell debris	
*Rana catesbeiana*		Sialic acid-binding lectin (SBLc)	Sialic acid	ZR-75-1		96 h (20 µg/mL)	Cell survival (120 h): <50% (MCF-7; ZR-75-1); >80% (MDA-MB-231; ZR-75-30)	[116]
				MCF-7		72 h (20 µg/mL)	↑ caspase-3 activity (MCF-7; ZR-75-1)	
				MDA-MB-231		96 h (0–40 µg/mL)	↓ER, ↓Bcl-2 (MCF-7)	
				ZR-75-30				
*Rana catesbeiana*		Sialic acid-binding lectin (SBLc)	Sialic acid	MCF-7		0–120 h (20 µg/mL)	Cell survival (120 h; SBLc): 13% (MCF-7); 31.3% (MCF-7/Bcl-xL)	[117]
*Rana catesbeiana*		Sialic acid-binding lectin (SBLc)	Sialic acid	Undifferentiated HL-60		120 h (2, 20 µg/mL)	Cell survival (120 h; 20 µM): 5.5% (undifferentiated)	[118]
				retinoic acid-differentiated HL-60		5–7–9 days of differentiation + 120 h (2, 20 µg/mL)	Cell viability (120 h; 20 µM; differentiated cells): 65% (5 days); 82.5% (7 and 9 days)	
						7 days of differentiation + 48, 96 h (20 µg/mL); 48, 96 h (20 µg/mL)	↑caspase-9, -3 and cleaved PARP (undifferentiated cells)	
							↑caspase-9 and -3 activity (undifferentiated cells)	
*Rana catesbeiana*		Sialic acid-binding lectin (SBLc)	Sialic acid	DBTRG		0–96 h (20 µg/mL)	Cell inhibition rate (96 h): ∼10% (RG2); ∼25% (GBM8401); ∼45% (DBTRG; GBM8901)	[119]
				GBM8901		0–96 h (2–50 µg/mL)	Cell inhibition rate (50 µg/mL; 96 h): ∼15% (RG2); ∼40% (DBTRG); ∼65% (GBM8901)	
				GBM8401		24, 72 h (50 µg/mL)	↑% of cells in sub-G1-phase (~30%; DBTRG)	
				RG2		72 h (50 µg/mL)	↑caspase-9 and -3 activity, not caspase-8 (DBTRG)	
					DBTRG cells subcutaneously injected in nude mice	a single injection (5 µg)	↓tumor size after 18 days of treatment	
*Rana catesbeiana*		Sialic acid-binding lectin (SBLc)	Sialic acid	HL60		120 h (20 µg/mL SBLc; 10 ng/mL IFN-𝛾)	Cell viability: SBLc + IFN-𝛾 < SBLc (MCF-7; SK-Hep-1); SBLc + IFN-𝛾 = SBLc (HL60)	[120]
				MCF-7		48, 96 h (20 µg/mL SBLc; 10 ng/mL IFN-𝛾)	HL60: ↑ caspase-3, -8, and -9 activity (SBLc + IFN-𝛾 = SBLc)	
				SK-Hep-1			MCF-7: ↑caspase-7 activity (SBLc + IFN-𝛾 > SBLc);	
							SK-Hep-1: caspase-3, -8, and 9 activity = control	
						48, 96 h (20 µg/mL SBLc + 10 ng/mL IFN-𝛾)	↑cleaved caspase-3 and PARP (HL60); ↑ cleaved caspase-7 and PARP (MCF-7)	
*Rana catesbeiana*		Sialic acid-binding lectin (SBLc)	Sialic acid	SK-Hep-1		0–96 h (20 µM SBLc + 10 ng/mL TNF-𝛼 or -𝛽)	Cell survival (96 h): ∼40% (J5); ∼50% (SK-Hep-1); ∼90% (HepG2)	[121]
				J5		0–120 h (20 µM SBLc + 10 ng/mL IFN-𝛾)	Cell survival (120 h, SK-Hep-1): 13.3% (SBLc+IFN-𝛾) vs 64.7% (SBLc)	
				HepG2			Cell survival (120 h, J5): 27.8% (SBLc+IFN-𝛾) vs 76.8% (SBLc)	
				BHK21			Cell survival (120 h, HepG2): 64.2% (SBLc+IFN-𝛾) vs 93.9% (SBLc)	
							Cell survival (120 h, BHK21): 91.52% (SBLc+IFN-𝛾) vs 96.67% (SBLc)	
*Rana catesbeiana*		Tobacco-derived his-HR Recombinant hHscFv–RC-RNase protein	Sialic acid	SMMC7721		24 h (0.7–3.5 nM)	IC50: 2 nM (SMMC7721); 2.4 nM (HepG2); 4.8 nM (DV145)	[122]
				HepG2			Cell inhibition rate (3.5 nM): ∼15% (HL-7702)	
				DV145				
				HL-7702				
*Rana japonica*		Sialic acid-binding lectin (SBLj)	Sialic acid	P388		48 h (0.1–5 µM)	GI501 (P388): 1.56 (SBLj)	[107]
				L1210			GI1001 (L1210): 1.56 (SBLj)	
**Fish**								
*Anguilla japonica*		Adenovirus FLAG (Ad.FLAG) Anguilla japonica lectin 1 (AJL1)	β-galactoside	Hep3B		48–120 h (50–100 MOI)	Cell viability (100MOI; 96h): ∼10% (SMMC-7721); ∼20% (Hep3B, BEL-7404, QSG-7701); ∼40% (Huh7); ∼60% (A549)	[123]
				BEL-7404		48 h (50 MOI)	Hep3B: AnnexinV^+^/propidium iodide^-^ 19.5% (Ad.FLAG-AJL1) vs 5.04% (Ad.FLAG)	
				Huh7			Hep3B: ↑cleaved PARP, Bcl-XL; ↓ procaspase-9, Bcl-2, XIAP	
				SMMC7721		48 h (50–100 MOI)	Hep3B: ↑PMRT5; ↓E2F-1	
				A549		48 h (50–100 MOI)	Hep3B: ↓ERK, ↓pERK, ↓p38	
				QSG-7701				
*Aristichthys nobilis*		Bighead carp gill lectin (GANL)		SMMC7721		24 h (0.5–64 µg/mL)	Cell inhibition rate (64 µg/mL): ∼0% (SMMC7721, BGC803); ∼20% (SKOV3, HepG2); ∼-30% (LoVo); ∼80% (HeLa)	[124]
				HepG2		72 h (0.5–64 µg/mL)	Cell survival (16 µg/mL): ∼115% (splenocytes)	
				SKOV3				
				HeLa				
				BGC803				
				LoVo				
				BALB/c mice splenocytes				
*Dicentrarchus labrax*		Adenovirus FLAG (Ad.FLAG)-*Dicentrarchus labrax* fucose-binding lectin (DlFBL)	Fucose	Hep3B		48–96 h (1–20 MOI)	Cell viability (20MOI; 96h; Ad.FLAG-DlFBL): ∼30% (Hep3B); ∼40% (BEL-7404, A549, SW480)	[61]
				BEL-7404		48 h (20 MOI)	Hep3B: Annexin V^+^/propidium iodide^-^ 21.5% (Ad.FLAG-DlFBL) vs 1.35% (Ad.FLAG)	
				A549			Hep3B: = cleaved PARP; ↓Bcl-2, ↓XIAP	
				SW480			Hep3B: ↓E2F-1	
*Dicentrarchus labrax*		*Dicentrarchus labrax* fucose-binding lectin (Ad-DlFBL) + coxsackie-adenovirus receptor (sCAR)-DlFBL	Fucose	K562/adr		48 h (5–30 MOI Ad-EGFP + 10 µg/mL sCAR-DlFBL)	Viral infection and replication: K562/ADR: 20% sCAR-DlFBL vs 3.19% Ad-EGFP; U87MG: 40.6% sCAR-DlFBL vs 23.1% Ad-EGFP	[57]
				U87MG		96 h (8.2 MOI Ad-DlFBL + 14-42 µg/mL sCAR-DlFBL)	Cell viability (U87MG): ∼20% (42 µg/mL Ad-DlFBL-sCAR-DlFBL); ∼50% (Ad-DlFBL)	
						48 h (8.2 MOI Ad-DlFBL + 42 µg/mL sCAR-DlFBL)	U87MG: AnnexinV^+^/propidium iodide^-^ 4.87% (Ad-DlFBL-sCAR-DlFBL) vs 7.91% (Ad-DlFBL)	
						48 h (8.2 MOI Ad-DlFBL + 42 µg/mL sCAR-DlFBL)	U87MG: ↑pERK (Ad-DlFBL-sCAR-DlFBL)	
*Oncorhynchus keta*		l-rhamnose-binding lectin (CSEL)	Rhamnose	Caco-2		24 h (1–100 µg/mL)	Cell viability: ~35% (Caco-2); no effects on DLD-1 or HCT-15	[125]
				DLD-1		24 h (1–100 µg/mL CSEL ± 0.1 M l-rhamnose or 2 µM PPMP)	Cell viability (Caco-2): ~35% (CSEL); ~90% (CSEL ± L-rhamnose or PPMP)	
				HCT-15		24 h (50–100 µg/mL)	DNA fragmentation in Caco-2	
							Annexin V^+^/propidium iodide^−^ (100 µg/mL; Caco-2): 32.8% (CSEL) vs 3.1% (control)	
*Oncorhynchus tshawytscha*		Rhamnose-binding roe chinook salmon lectin (CSRL)	Rhamnose	*MCF-7*		24–48 h (3.9–250 μM)	IC50 (24–48 h): 93–45 µM (HepG2); 220–68 µM (MCF-7);	[126]
				*Hep G2*		48 h (68 µM)	WRL68: no effect	
				*WRL68*		24 h (0.625–20 µM)	NO production at 0.62 µM (at 20 µM CSRL)	
				*Mouse peritoneal macrophages*				
*Silurus asotus*		Rhamnose-binding lectin (SAL)	Rhamnose	*Raji*		5–30 min (2.5–10 μg/mL)	Raji (30 min; 10 µg/mL): ↑ AnnexinV+/propidium iodide- (16.7% SAL vs 4.31% control) and AnnexinV+/ propidium iodide+ (77.25% SAL vs 4.79% control)	[127]
				*K562*			Raji (30 min; 10 µg/mL): ↑ 30-fold shrunken cell population	
				*K562/DXR*			K562; K562/DXR: no effects	
						30 min (10 μg/mL SAL + 4 µM CsA)	Raji: ↓ necrotic cells (58.03% SAL + CsA vs 77.92% SAL)	
*Silurus asotus*	Rhamnose-binding lectin (SAL)		Rhamnose	Raji		24–120 h (0–100 μg/mL)	Cell viability: no effects	[128]
						24–48 h (100 µg/mL)	Cell proliferation: block at 50 µg/mL	
						72 h (100 µg/mL) + 48 h SAL-free medium	Cell proliferation: restored	
						24 h (100 µg/mL)	↑% of cells in G0/G1-phase (20%); ↓% of cells in S-phase (20%)	
						12–24 h (100 µg/mL SAL ± 20 mM Saccharide)	↓ CDK4, C-MYC (40%), CCND3 (30%); = CDK2; ↑ p21 (130%), p27 (70%); + saccharide reverts effects (except for p27)	
						12–24 h (100 µg/mL)	↓CDK4, ↓c-Myc, ↓cyclin D3, ↑ p21, ↑p27	
						0.5–24 h (100 µg/mL)	↑ GTP-Ras, pMEK, pERK; = pp38 and cJNK	
						(100 µg/mL) in A4GALT ^4^siRNA Raji cells	↓pMEK, ↓pERK induced by SAL	
						2 h (10 µM U0126) ± 12 h (100 µg/mL SAL)	↓p21, ↓pERK induced by SAL (cell-cycle arrest depending on Ras-MEK-ERK pathway)	
							↑cell proliferation rate	

^1^ Concentration that inhibits the growth of cells by 50%; ^2^ concentration that inhibits the growth of cells by 100%; ^3^ combination index; ^4^ A4GALT: 𝛼-1,4-galactosyltransferase.

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
