# Peer review of "Antitumor Potential of Marine and Freshwater Lectins"

_marinedrugs, 2019, doi:10.3390/md18010011_

Round 1

Reviewer 1 Report

Dear Editor and authors, 

The review addresses a very important and not well-known topic, the lectins from the sea and freshwater. 

Introduction is very nice as it explains cancer and lectins even from non-experts readers in the field. 

The tables provide a clear and brief information of the activity of each lectin. 

Figures 1 and 2 are very attractive and informative. 

Here my suggestions:

General suggestions

Classification of Lectins is of utmost importance which is not mentioned in the introductory part of this review. If possible, insert in a tabular form specifying the types of lectins, cells of interaction, ligand with which it binds.

If possible, kindly mention what percentage of lectins in market are marine/ freshwater/ terrestrial based, so as to compare the significance of Marine lectins.

For the lectins, as much as possible in the text include the structural information. 

The review article will be much more attractive to read if pictorial representation is available which would suffice the need of reader.

It is very important than in tables 1, 2 and 3, a column indicating “Recognition glycans” for each lectin should be included (in case it is not discovered, just leave it blank)

In some cases, the order of the lectins in the tables does not follow the same order as the explanation. For the best understanding of the reader, I suggest following the same order. 

As this review is more inclined towards the pharmacological aspect, if possible, provide the adverse effects of every lectin, if any.

There are several methods to express proteins, is there any study which gives idea about the bulk production of lectins?

Considering the pharmacological aspect, is there study or clinical trial mentioning the synergistic effect of using two or more lectins, if any, please provide.

Minor suggestions

Page 1, line 46: I am not sure, but according to the next sentence I think that maybe the authors wanted to write “many sessile” instead of “many non-sessile”

The authors explain clear about tumor processes and why glycans on cancer cells are different than on non-transformed cells. After, there is a clear explanation about what are lectins and recognition pattern. After, the authors explain the lectins and which molecules they bind. However, there is no an introduction of the different glycans in cancer cells. I suggest to include an introduction of the over expressed and novel glycans expressed by cancer cells. Then, when we read about lectins and recognition molecules, the reader can understand why those lectins are important as antitumoral agents. 

Page 2, line 81: “hypoxia, and acidity” should be changed to "hypoxia and acidity”

Page 3, lines 121-123: For making it clear to the reader, this sentences should be reformulated: “In particular, lectins bind to a 121 specific sequence of monosaccharide moieties within glycosylated proteins, lipids, and glycans that 122 are on cell surface [19], thus enabling cell-cell interaction.”

Page 3, Section 2 (2. lectins): In the paragraph explaining about the lectins recognition, for the reader is not clear if lectins are secreted proteins or lectins attached to other cells. Then, specification about type of cell that secrete those lectins should be included (dendritic cell, macrophages…).

Page 4, Table 1:  “Cellular and Molecular Target” column. The list should follow a criteria, I suggest two possibilities: 1) in same order as “origin” column; 2) cell growth inhibition order. For instance, if we follow the same order as “origin” for the MCF-7 cells, orders should be: “ % of cell growth inhibition in MCF-7 cells: 1.78 (A. spicifera); 9.10 (A. nana); 12.25 (H. divaricata); 28.54 (G. repens); 15.53 (N. punctatum)“  (And same criteria should be applied from now on for HeLa cells and for the other tables).

Page 4, Table 1: The section “Eucheuma serra” from reference 33. Information is confused. Please, remake this section for a clear comprehension. 

Idem from the information given from reference 35 (page 5, Table 1). 

Page 5, line 151: Mention the anomer with which ESA binds with Mannose.

Page 6, lines 159-166: If possible, mention any study where these lectins bind to normal high mannose containing cell/tissues.

Page 6, lines 167-190: Exemplify the structural specificity of lectin interacting with sugar, i.e., which part of lectin interacts with sugar.

Page 6, lines 195/205: Sfls induces / Sfls upregulate ? Singular o plural

Page 7, line 211: Specify the conformation of sugar to which UPL1 binds.

Page 7, line 211: “N-acetylglucosamine” should be changed to “N-acetylglucosamine”

Table 2: h (2.5-20 μg/mL), maybe 1h?

Page 7, lines 231- 232: Give examples of other methods used to understand the presence of lectins in samples.

Page 7, lines 240-247: Outline the other methods used to purify lectins.

Page 7, line 249: “belong” should be changed by “belonging”.

Page 12, lines 275-284: Provide, if any, the effect of CGL on normal cells.

Page 12, line 310-311: To avoid confusion, the sentence “…lectin has been found in mice to abolish” should be changed by “… lectin has been injected on mice and it was found a …”

Page 13, line 313: “ruote of administration” should be changed to “route of administration”

Page 13, lines 322- 329: Specify the carbohydrate it binds.

Page 14, lines 381: "N-acetylhexosamine-binding lectin” should be changed to “N-acetylhexosamine-binding lectin “

Page 22, line 461: Mention which motif of SBLc binds with Sialic acid, if any study provided.

Page 33, line 355: “the same sensitivityy” should be changed by “the same sensitivity” 

Page 26, Figure 2: If possible, it would be great to include the name of the organism  at the bottom of each image. 

Sincerely yours, 

Reviewer

Author Response

The authors of this manuscript express their sincere thanks to the reviewer for the critical assessment of our work. The authors have acted upon the recommendations of the reviewer which have resulted in a significant enhancement of the quality of this manuscript. All modifications incorporated in the manuscript are highlighted using red color font. A “point-by-point” response to the reviewer’s comments is outlined below.

General comments:

The review addresses a very important and not well-known topic, the lectins from the sea and freshwater.

Introduction is very nice as it explains cancer and lectins even from non-experts readers in the field.

The tables provide a clear and brief information of the activity of each lectin.

Figures 1 and 2 are very attractive and informative.

Response:

We greatly appreciate the reviewe’s generous comments about the quality of our manuscript.

Specific comments:

Major comments:

Comment 1:

Classification of Lectins is of utmost importance which is not mentioned in the introductory part of this review. If possible, insert in a tabular form specifying the types of lectins, cells of interaction, ligand with which it binds.

Response:

Since we could not find a univocal way to classify lectins and not for all lectins has been described the category of belonging, we have inserted a figure (Figure 1) and a small paragraph in the introduction about the common characteristics that allow general lectins classification (lines 118-120). Cells of interaction is already a piece of information in Tables 1, 2 and 3, and in the same tables we have also added the ligand with which they bind.

Comment 2:

If possible, kindly mention what percentage of lectins in market are marine/ freshwater/ terrestrial based, so as to compare the significance of Marine lectins.

Response:

This is, indeed, a thought-provoking comment. As far as we know, currently there are no lectins in market as antitumor agents.

Comment 3:

For the lectins, as much as possible in the text include the structural information.

Response:

We have included structural information for most lectins throughout the revised manuscript and added a figure representing similarities of primary structure of different groups of lectins (Figure 2).

Comment 4:

The review article will be much more attractive to read if pictorial representation is available which would suffice the need of reader.

Response:

We are in absolute agreement with the reviewer. We have added one figure about lectin classification (Figure 1) and the comparison of amino acid sequencing of different group of lectins (Figure 2). We sincerely believe these new figures have increased the didactic appeal of our review.

Comment 5:

It is very important than in tables 1, 2 and 3, a column indicating “Recognition glycans” for each lectin should be included (in case it is not discovered, just leave it blank).

Response:

In all tables, we have included the “Recognition glycans” column.

Comment 6:

In some cases, the order of the lectins in the tables does not follow the same order as the explanation. For the best understanding of the reader, I suggest following the same order.

Response:

We modified the tables and now they are organised by category and following the alphabetical order. In this way, readers can easily find the information they need.

Comment 7:

As this review is more inclined towards the pharmacological aspect, if possible, provide the adverse effects of every lectin, if any.

Response:

We reported all the toxic effects in the text. What emerges is that in the experimental conditions (both in vitro and in vivo) lectins have an extremely safe toxicological profile. However, in the conclusion, we have added a small paragraph about general lectin toxicity (lines 764-770).

Comment 8:

There are several methods to express proteins, is there any study which gives idea about the bulk production of lectins? 

Response:

One of the hardest challenges about lectins is exactly bulk production. This problem is the result of the high costs and complexity of purification and characterization of these proteins. Moreover, their therapeutic potential has been investigated not for long time. For these reasons, the easiest and most efficient way to produce lectins is exploiting heterologous systems, such as bacteria and yeast.

We have added two small sentences and a fitting reference in the conclusion (lines 780-781 and 783-785).

Comment 9:

Considering the pharmacological aspect, is there study or clinical trial mentioning the synergistic effect of using two or more lectins, if any, please provide.

Response:

We could not find any study or clinical trial mentioning the synergistic effect of multiple lectins.

Minor suggestions

Comment 1:

Page 1, line 46: I am not sure, but according to the next sentence I think that maybe the authors wanted to write “many sessile” instead of “many non-sessile”.

Response:

As suggested, we have rectified the mistake (line 47).

Comment 2:

The authors explain clear about tumor processes and why glycans on cancer cells are different than on non-transformed cells. After, there is a clear explanation about what are lectins and recognition pattern. After, the authors explain the lectins and which molecules they bind. However, there is no an introduction of the different glycans in cancer cells. I suggest to include an introduction of the over expressed and novel glycans expressed by cancer cells. Then, when we read about lectins and recognition molecules, the reader can understand why those lectins are important as antitumoral agents.

Response:

To fill the gap that reviewer 1 pointed out, we have added a small explanatory paragraph about the glycosylation pattern in cancer cells (lines 139-153).

Comment 3:

Page 2, line 81: “hypoxia, and acidity” should be changed to "hypoxia and acidity”.

Response:

As suggested, we have rectified the mistake (line 82).

Comment 4:

Page 3, lines 121-123: For making it clear to the reader, this sentences should be reformulated: “In particular, lectins bind to a 121 specific sequence of monosaccharide moieties within glycosylated proteins, lipids, and glycans that 122 are on cell surface [19], thus enabling cell-cell interaction.”

Response:

We have reformulated the sentence (132-134).

Comment 5:

Page 3, Section 2 (2. lectins): In the paragraph explaining about the lectins recognition, for the reader is not clear if lectins are secreted proteins or lectins attached to other cells. Then, specification about type of cell that secrete those lectins should be included (dendritic cell, macrophages…).

Response:

Almost all types of cells express lectins, including antigen presenting cells as Reviewer 1 suggested, and, depending on the type of lectin, they can be exposed on the cellular membrane or secreted into the extracellular matrix. We have specified this information in the text (lines 130-132).

Comment 6:

Page 4, Table 1:  “Cellular and Molecular Target” column. The list should follow a criteria, I suggest two possibilities: 1) in same order as “origin” column; 2) cell growth inhibition order. For instance, if we follow the same order as “origin” for the MCF-7 cells, orders should be: “ % of cell growth inhibition in MCF-7 cells: 1.78 (A. spicifera); 9.10 (A. nana); 12.25 (H. divaricata); 28.54 (G. repens); 15.53 (N. punctatum)“  (And same criteria should be applied from now on for HeLa cells and for the other tables).

Response:

We have arranged the tables following “cell growth inhibition order”.

Comment 7:

Page 4, Table 1: The section “Eucheuma serra” from reference 33. Information is confused. Please, remake this section for a clear comprehension.

Response:

We have clarified the concept.

Comment 8:

Idem from the information given from reference 35 (page 5, Table 1).

Response:

We have clarified the concept.

Comment 9:

Page 5, line 151: Mention the anomer with which ESA binds with Mannose.

Response:

As suggested by the reviewer, we have inserted this information in the text (185-189).

Comment 10:

Page 6, lines 159-166: If possible, mention any study where these lectins bind to normal high mannose containing cell/tissues.

Response:

Unfortunately, we could not find any information about that. The binding to mannose residues has been demonstrated only on tumor cell lines.

Comment 11:

Page 6, lines 167-190: Exemplify the structural specificity of lectin interacting with sugar, i.e., which part of lectin interacts with sugar.

Response:

The part of lectin interacting with the sugar is the same of free ESA, thus we added this information in lines 180-184.

Comment 12:

Page 6, lines 195/205: Sfls induces / Sfls upregulate ? Singular o plural

Response:

It is plural; we have corrected the mistake (line 245).

Comment 13:

Page 7, line 211: Specify the conformation of sugar to which UPL1 binds.

Response:

As suggested by the reviewer, we have specified the conformation of sugar to which UPL1 binds (line 264).

Comment 14:

Page 7, line 211: “N-acetylglucosamine” should be changed to “N-acetylglucosamine”

Response:

The correction has been made.

Comment 15:

Table 2: h (2.5-20 μg/mL), maybe 1h?

Response:

24h is the correct treatment time; we have added it into the table.

Comment 16:

Page 7, lines 231- 232: Give examples of other methods used to understand the presence of lectins in samples.

Response:

As suggested by the reviewer, we have added other examples of methods used to understand the presence of lectins in samples (lines 282-283).

Comment 17:

Page 7, lines 240-247: Outline the other methods used to purify lectins.

Response:

We have added that information (lines 292-294).

Comment 18:

Page 7, line 249: “belong” should be changed by “belonging”.

Response:

As suggested by the reviewer, we have corrected the mistake (line 309).

Comment 19:

Page 12, lines 275-284: Provide, if any, the effect of CGL on normal cells.

Response:

To our knowledge, no study was performed assessing the effects of CGL on normal cells.

Comment 20:

Page 12, line 310-311: To avoid confusion, the sentence “…lectin has been found in mice to abolish” should be changed by “… lectin has been injected on mice and it was found a …”

Response:

As suggested by the reviewer, we have corrected the sentence (lines 376-377).

Comment 21:

Page 13, line 313: “ruote of administration” should be changed to “route of administration”.

Response:

As suggested by the reviewer, we have corrected the mistake (line 379).

Comment 22:

Page 13, lines 322- 329: Specify the carbohydrate it binds.

Response:

As suggested by the reviewer, we have added the missing information (line 393).

Comment 23:

Page 14, lines 381: "N-acetylhexosamine-binding lectin” should be changed to “N-acetylhexosamine-binding lectin “

Response:

The correction has been made (line 462).

Comment 24:

Page 22, line 461: Mention which motif of SBLc binds with Sialic acid, if any study provided.

Response:

We have added the few information available (lines 555-561).

Comment 25:

Page 33, line 355: “the same sensitivityy” should be changed by “the same sensitivity”

Response:

As suggested by the reviewer, we have corrected the mistake (line 429).

Comment 26:

Page 26, Figure 2: If possible, it would be great to include the name of the organism at the bottom of each image.

Response:

This is an excellent suggestion. We have included the name of each organism at the bottom of the Figure 4.

Additionally,

The reference list has been modified as we have added several new references. Special attention is given to conform to the order of references and bibliographic style of the journal. The entire manuscript has been thoroughly checked and edited to ensure uniform style, organization and quality.

On behalf of my co-authors, I once again express my sincere thanks to the erudite reviewer for the valuable suggestions and constructive input to improve the quality of our manuscript.

Reviewer 2 Report

I read with attention the review entitled: Antitumor potential of marine and freshwater lectins. This review presents a considerable number of marine lectins with different research work about them.

This paper is well written and well documented and very interesting.

I recommend publishing with minor modifications.

Just minor remarks

Line 313 : replace ruote by route (typing error)

Line 326 : explain the acronym oncolytic VV

Line 516: the sentence is unclear “The cause of ER and PR downregulation of the could…

Author Response

The authors of this manuscript express their sincere thanks to the reviewer for the critical assessment of our work. The authors have acted upon the recommendations of the reviewer which have resulted in a significant enhancement of the quality of this manuscript. All modifications incorporated in the manuscript are highlighted using red color font. A “point-by-point” response to the reviewer’s comments is outlined below.

General comments:

I read with attention the review entitled: Antitumor potential of marine and freshwater lectins. This review presents a considerable number of marine lectins with different research work about them.

This paper is well written and well documented and very interesting.

I recommend publishing with minor modifications.

Response:

We are grateful to the reviewer for evaluation of our work with positive comments.

Specific comments:

Comment 1:

Line 313 : replace ruote by route (typing error)

Response:

As suggested by the reviewer, we have corrected the mistake (line 379).

Comment 2:

Lines 326 : explain the acronym oncolytic VV

Response:

On lines 158 and 392, we explained the acronym oncolytic VV.

Comment 3:

Line 516: the sentence is unclear “The cause of ER and PR downregulation of the could…

Response:

As suggested by the reviewer, we have corrected the sentence (609-610).

Additionally,

The reference list has been modified as we have added several new references. Special attention is given to conform to the order of references and bibliographic style of the journal. The entire manuscript has been thoroughly checked and edited to ensure uniform style, organization and quality.

On behalf of my co-authors, I once again express my sincere thanks to the erudite reviewer for the valuable suggestions and constructive input to improve the quality of our manuscript.